# Time-programmable drug dosing allows the manipulation, suppression and reversal of antibiotic drug resistance *in vitro*

Mari Yoshida[1], Sabrina Galiñanes Reyes[1], Soichiro Tsuda[1], Takaaki Horinouchi[2], Chikara Furusawa[2,3]
& Leroy Cronin[1]

Multi-drug strategies have been attempted to prolong the efficacy of existing antibiotics, but with limited success. Here we show that the evolution of multi-drug-resistant *Escherichia coli* can be manipulated *in vitro* by administering pairs of antibiotics and switching between them in ON/OFF manner. Using a multiplexed cell culture system, we find that switching between certain combinations of antibiotics completely suppresses the development of resistance to one of the antibiotics. Using this data, we develop a simple deterministic model, which allows us to predict the fate of multi-drug evolution in this system. Furthermore, we are able to reverse established drug resistance based on the model prediction by modulating antibiotic selection stresses. Our results support the idea that the development of antibiotic resistance may be potentially controlled via continuous switching of drugs.

[1] WestCHEM, School of Chemistry, The University of Glasgow, Glasgow G12 8QQ, UK. [2] Quantitative Biology Center, RIKEN, 6-2-3 Furuedai, Suita, Osaka 565-0874, Japan. [3] Department of Physics, University of Tokyo, 7-3-1 Hongo, Bunkyo-ku, Tokyo 113-0033, Japan. Correspondence and requests for materials should be addressed to S.T. (email: Soichiro.Tsuda@glasgow.ac.uk) or to L.C. (Lee.Cronin@glasgow.ac.uk).

The emergence of multi-drug-resistant bacteria, or 'superbugs', poses an imminent threat to our society and has been accelerated by a number of factors, such as the overuse and misuse of existing antibiotics as well as diminishing antibiotic pipelines[1,2]. The use of a combination of drugs as a multi-drug strategy, such as combination therapy and antibiotic cycling, has been proposed and used to cope with the current situation. To date, a wide range of studies have been conducted to elucidate the effects of single as well as multiple drugs on bacterial evolution. *In vitro* pharmacokinetic/pharmacodynamic studies have identified optimal dosing regimens using multiple drugs that can effectively suppress bacterial growth and prevent the emergence of drug-resistant mutants[3–6]. Laboratory evolution studies utilising whole-genome sequencing[7–12] and transcriptome analysis[13,14] have been applied to investigate longer-term dynamics of bacterial adaptation to stressful drug environments, and provide insight to the complex relationships between drug resistance, and genetic alterations and gene expression changes. Theoretical modelling has been employed to understand bacterial evolution from a systems perspective. Various models including phamacodynamics[15,16], population genetics[17,18] and population dynamics[19,20] models have been developed to examine results obtained from experimental studies. Antibiotic cycling has also been studied in clinical settings for over 30 years, particularly cycling of aminoglycosides was widely studied due to increasing drug resistance mediated by plasmids carrying aminoglycoside enzymes[21,22]. Cycling in the intensive care unit was also actively investigated because infections with drug-resistant bacteria can be lethal if treatment fails[23].

Despite a number of clinical, laboratory and theoretical studies with various antibiotic combinations[20,24–28], they showed mixed results partly because of a lack of standard procedures to perform experiments. However, recent studies suggest that exploitation of collateral sensitivity, in which bacterial strains resistant to an antibiotic exhibit increased susceptibility to other antibiotics[12,13,29], may be key to the suppression or reversal of the evolution of bacterial drug resistance[30]. Thus temporally modulated use of different antibiotics is a promising candidate method for effective multi-drug dosing because switching of antibiotics can selectively perish resistant strains due to fitness costs associated with drug resistance while allowing susceptible strains to overgrow. So far several laboratory evolution studies based on collateral sensitivity have demonstrated the effectiveness of antibiotic cycling with certain drug combinations and dosing regimens[9,10,31,32]. However, an exhaustive experimental investigation of effective antibiotic cycling combinations for suppressing the emergence of bacterial drug resistance is yet to be conducted.

In this work, we investigate effective combinations of antibiotics that suppress the development of drug resistance under cycled stress and how the multi-drug evolution can be related to evolution under single antibiotic condition. We find unique evolutionary patterns (that is, temporal development of bacterial drug resistance) in which the emergence of drug resistance to one of the cycled antibiotics is completely suppressed. A simple mathematical model is derived from the obtained data, which allows us to predict the fate of multi-drug evolution from evolutionary patterns under single drug conditions. By combining the experimental data and model predictions, we then demonstrate that reversing evolved multi-drug resistance is possible by temporally modulating antibiotic stresses.

## Results

**Experiment design and platform.** This study aims to gain insights into bacterial evolution under fluctuating antibiotic stress by the design of a series of long-term (24 days) laboratory evolution experiments. Experiments were designed to expose bacterial cultures to single antibiotic stress and a range of configurable alternating antibiotic stresses. The drug type and the concentration of the drug applied were either switched (on/off) or oscillated over a range of variable time periods (Fig. 1a). To achieve this type of programmable environmental drug condition we designed and constructed an automated morbidostat platform[7], application of which enabled assessment of the development of antibiotic resistance (measured as changes in susceptibility) as a function of multi-drug composition, relative concentrations and time (Supplementary Fig. 1). In the morbidostat, the antibiotic concentration was automatically controlled by a custom algorithm[7]; when a bacterial culture reaches a certain density, 1 ml of a growth medium containing an antibiotic was administered into 12 ml of a bacterial culture, while a medium containing no antibiotic was added at low cell density (Methods). The antibiotic concentration in the administered growth medium was at least 10 times higher than the minimum inhibitory concentration (MIC) of a wild-type strain. Typically, antibiotics were added at least a few times once the cell density exceeds a threshold level, and thus the antibiotic concentration was increased to the range of drug concentrations that select for drug-resistant mutants (so-called 'mutant-selection window'[33]). The use of the automated cell culture system was important not only in delivering programmable dosing, but the morbidostat system enabled maintenance of bacterial populations in exponential growth phase. Thus the platform was expected to provide good phenotypic reproducibility between parallel experiments[7] whilst enabling genetically diverse populations to co-exist[34,35]. Note that we here defined a wild-type *Escherichia coli* MG1655 strain as a susceptible strain, in contrast to the clinical definition in which a strain can be treated with an antibiotic at the recommended dosage. Resistant strains were defined as any (evolved) bacterial samples whose MICs exceed that of the wild-type strain.

**Bacterial evolution under alternating antibiotic stresses.** Starting with an isogenic *E. coli* strain, a series of 24-day experiments in the morbidostat system was performed using two alternating antibiotic stresses (Fig. 1a), as well as 12-day experiments under a single antibiotic stress. Antibiotics were chosen from different classes so that they have different targets (Table 1), which minimize the risk of developing potential cross resistance between antibiotics[12,13,29]. To assess the reproducibility of the evolutionary trajectories, experiments were duplicated for each antibiotic cycling pair and triplicated for each single antibiotic. Experiments with reversed order of antibiotic cycling were also performed in duplicate to evaluate the effect of order of cycling. These experiments revealed that the bacterial populations exhibited two evolutionary patterns in the development of drug resistance under the cycled conditions as shown in Fig. 1. The first group developed the resistance to only one of the cycled antibiotics, whereas the second group became resistant to both antibiotics. We refer to these antibiotic-resistant conditions of the bacterial populations as a 'single drug' and 'multi-drug' resistant state, respectively. Within the second category, two different patterns were observed: An oscillation[30] with a gradual increase of baseline (Fig. 1g) and a lowered maximum resistance (Fig. 1i). In the former case, the rates of adaptation to chloramphenicol (CHL) and nitrofurantoin (NIT) were both delayed (the steepness of fitted logistic curves $k = 0.4 \pm 0.03$ and $0.19 \pm 0.01$ day$^{-1}$, mean ± s.e.m., respectively) compared to the case of single antibiotic stress (Fig. 1f, $k = 0.79 \pm 0.13$ and $0.55 \pm 0.05$ day$^{-1}$, respectively). In the latter case, the drug resistance was developed rapidly and at a similar rate to the single antibiotic case

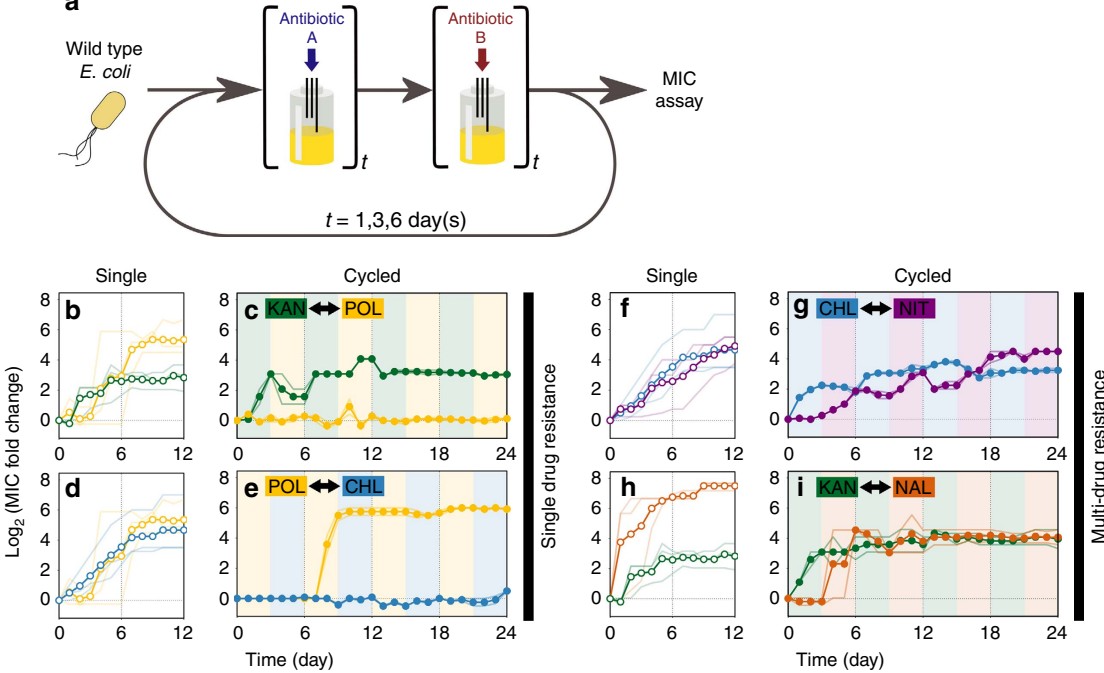

**Figure 1 | Evolution of antibiotic resistance under antibiotic cycling condition.** (**a**) Experimental design for bacterial evolution under cyclic exposure to two antibiotics. Wild-type *E. coli* was cultured in the morbidostat system with dynamic antibiotic concentrations. The Every *t* days (where *t* = 1, 3 and 6), the drugs were switched in an alternating manner and the cell cultures were sampled daily and stored at − 80 °C. After a total of 12 or 24 days the MICs of the collected samples were determined. (**b**) Evolutionary trajectories of KAN and POL resistance (green and yellow curves, respectively) under single antibiotic condition. Trajectories of parallel experiments were shown as pale lines and the average of parallel experiments was in solid line with empty circles. The resistance levels were measured as MIC fold change relative to the wild-type, calculated by $\log_2(MIC_i/MIC_{WT})$, where $MIC_i$ and $MIC_{WT}$ are the MICs of the *i*-th day sample and wild-type *E. coli*, respectively. (**c**) Trajectories of KAN and POL resistance when cycled with 3 day interval. Pale lines indicate trajectories of parallel experiments and the average was shown as solid line with filled circles. The pale coloured background indicates the antibiotics used. (**d,e**) evolutionary trajectories of CHL and POL resistance (blue and yellow) under single antibiotic condition and cycling with 3 day interval, respectively. Similarly, for (**f,g**) CHL and NIT resistance (blue and purple), and (**h**) and (**i**) KAN and NAL resistance (green and orange). Samples sizes are *n* = 3 for single antibiotic condition and *n* = 2 for antibiotic cycling (biological duplicates).

$k = 1.7 \pm 0.12$ and $31.9 \pm 3.12$ day$^{-1}$ for kanamycin (KAN) and nalidixic acid (NAL), respectively, for cycling case and $k = 7.11 \pm 6.01$ and $5.06 \pm 1.65$ day$^{-1}$ for single antibiotic case. However, the maximum level of NAL resistance was significantly lowered ($P < 0.001$, Welch's two sample *t*-test. Supplementary Fig. 4a), while that of KAN resistance was comparable to the single antibiotic case. Evolutionary trajectories were reproducible between parallel experiments and the evolution experiments with reversed cycling order of antibiotics showed similar evolutionary patterns (Supplementary Fig. 2a).

While the development of multi-drug resistance was expected, the results of cycling with polymyxin B (POL) shown in Fig. 1c,e were surprising. This is because, depending on the antibiotic combinations, either the development of POL resistance or the counteracting antibiotic was suppressed. As a result, the bacterial population was resistant to only one antibiotic. In contrast, the resistance was developed in all cases when cultured under single antibiotic stress (Fig. 1b,d,f,h). To gain a better understanding of this behaviour, the bacterial population was challenged with various antibiotic combinations (Supplementary Fig. 3). Here, the antibiotics were cycled over an interval of 1 day because shorter intervals tended to delay the development of the antibiotic resistance (Supplementary Fig. 4b–i). The results showed that cycling with POL often drove the bacterial population to a single drug-resistant state, or kept the population in that state for a long period (Supplementary Fig. 3a). In contrast, cycling with the other antibiotics ended up in a multi-drug-resistant state in all the cases (Supplementary Fig. 3b).

**Bacterial evolution under single antibiotic stress**. We hypothesized that the single drug-resistant state with POL might be related to the evolutionary patterns of resistance under single antibiotic condition. Indeed, the development of POL resistance under single antibiotic stress exhibited a unique evolutionary pattern with a relatively long 'silent phase' where the population does not develop drug resistance (up to 6 days) followed by sharp increase as shown in Fig. 1b and Supplementary Fig. 5. As such, the evolutionary patterns of the other antibiotic resistance profiles can be categorized into two types: (1) The rapid development of NAL, KAN and rifampicin resistance increasing to the maximum level with little or no silent phase, and (2) a gradual increase in resistance for CHL, NIT, ampicillin and tetracycline over time[7,8]. Cycling POL with the antibiotics in the former category tends to suppress the resistance to POL, while populations developed POL resistance at a relatively early stage when cycled with the antibiotics in the latter category.

**Collateral sensitivity profile**. We also investigated the cross resistance and collateral sensitivity profiles[12,13,29] using the final day samples from the evolution experiments under a single antibiotic condition (Fig. 2). This showed that POL resistant strains showed collateral sensitivity to many antibiotics, indicating that having POL resistance tends to make bacterial cells more susceptible to other antibiotics than wild-type strains. On the other hand, strains that have already evolved resistance to the other antibiotics often developed cross resistance to the

**Table 1 | List of antibiotics used in this study.**

| Name | Abbreviation | Class | Target | Type | Solvent |
|---|---|---|---|---|---|
| Polymyxin B | POL | Polymyxin | Lipopolysaccharide | Bactericidal | Ethanol |
| Chloramphenicol | CHL | Amphenicol | Protein synthesis, 50S | Bacteriostatic | Water |
| Nitrofurantoin | NIT | Nitrofuran | Multiple mechanisms | Bactericidal | DMF |
| Nalidixic acid | NAL | Quinolone | DNA gyrase | Bactericidal | Water |
| Kanamycin | KAN | Aminoglycoside | Protein synthesis, 30S | Bactericidal | Water |
| Ampicillin | AMP | β-Lactam | Cell wall synthesis | Bactericidal | Water |
| Rifampicin | RIF | Rifamycin | RNA polymerase | Bactericidal | DMF |
| Tetracycline | TET | Tetracycline | Protein synthesis, 30S | Bacteriostatic | Ethanol |

DMF, dimethylformamide.

**Figure 2 | The collateral sensitivity and cross resistance profile of resistant strains.** MICs of the final day samples from the evolution experiments under single antibiotic conditions were tested with all eight antibiotics. Blue and red grids indicate that the antibiotic pair is collateral sensitive and cross resistance, respectively. The numbers in the grids are $\log_2$-transformed MIC fold change. The values shown in the matrix were rounded. See Supplementary Table 2 for the original data.

other antibiotics or remained neutral. While these results were in general consistent with previous studies on collateral sensitivity (Supplementary Fig. 12)[29,36], some collateral sensitivity profiles (for example, KAN) were less evident compared to previous cases. This may be due to different selection strengths during evolution between previous and our cases because the collateral sensitivity/cross resistance profile is known to be dependent on the selection strength[36].

**Possible mechanisms of drug resistance suppression.** The multi-drug and single drug evolution experiments demonstrated various evolutionary patterns depending on the types and combinations of antibiotics. While antibiotic cycling strategies that eliminated bacteria with sublethal antibiotic dosages have been reported[10], our case maintained exponential growth yet suppressed the emergence of antibiotic resistance (Fig. 1c,e). These various patterns of evolutionary trajectories make a stark contrast with a previous laboratory evolution study on antibiotic cycling where monotonic increase in drug resistance was consistently observed[9]. To provide possible explanations for the diverse evolutionary patterns, we first consider the growth conditions in the previous and our studies.

With the serial transfer method used in the previous study[9], a bacterial population experiences exponential and stationary phases during overnight culture. While the growth rate is a main factor for resistance evolution in exponential phase, other factors such as cell–cell interaction and resource competition would come into play in stationary phase. Evolution using the serial transfer method would thus be slower than the case when

the exponential phase was maintained (that is, morbidostat) because the bacteria need to adapt to both conditions in exponential and stationary phases. Antibiotic cycling would further slow the adaptation process because of collateral sensitivity, additional fitness costs by resistance to a second antibiotic and other factors[9].

In contrast, the rate of resistance evolution in the morbidostat is primarily determined by the growth rate, in particular the growth rate of a new mutant relative to that of the population[8]. This would explain the evolutionary patterns under single antibiotic stress as the growth rate (that is, fitness) of an antibiotic-resistant mutant may vary depending on the antibiotics used. For example, if only one single mutation is required to become strongly resistant to an antibiotic, such resistant mutant will quickly sweep through the population in the presence of the antibiotic and resistance at the population level rises sharply. On the other hand, if a fitness gain by a mutation is relatively small, the evolution will slowly increase. These mechanisms may explain the rapidly or gradually increasing evolutionary patterns (for example, KAN or CHL, respectively, Supplementary Fig. 5).

The evolution under single POL stress showed a unique pattern with a relatively long silent phase that was not observed in the other antibiotics tested. A simple explanation for the pattern could be to assume that multiple mutations are required for POL resistance. However, it seems implausible because the reproducible evolutionary trajectories of POL resistance (for example, Fig. 1e) cannot be explained considering the inherent stochasticity of genetic mutation. In addition, as described below, the whole-genome sequence data did not support the explanation (Supplementary Table 1). Instead, we speculate a two-step adaptation mechanism of POL resistance because it has been known that there are non-genetic resistance mechanisms to POL, such as Lipid A modification[37] and heteroresistance[38]. Recent studies indicated that heteroresistance to POL was achieved by upregulations of putrescine synthesis and YceI protein without any genetic mutations. In addition, release of the molecules from highly resistant subpopulations of heteroresistant bacteria protects less resistant bacteria from POL and other antibiotic stresses[39,40]. This evidence suggests that the E.coli populations in this study might have used the non-genetic mechanisms to cope with POL stress at the initial stage of evolution when the drug concentration was low. This could then have switched to genetic mechanisms, that is, beneficial mutations conferring POL resistance as the drug concentration increased, which may explain the step-like evolutionary pattern of POL resistance.

Based on the single drug evolutionary patterns, we then consider the evolutionary patterns of the antibiotic cycling, in particular KAN-POL (Fig. 1c) and POL-CHL (Fig. 1e) cycling. In the former case, a bacterial population may first acquire resistance to KAN as it was demonstrated to occur rapidly during the initial

stage of evolution experiments (Fig. 1b). Considering that having KAN resistance generally incur fitness costs, this may then have significantly delayed evolution to acquire additional resistance to POL by extending the silent phase, especially because POL resistance comes with large fitness costs[37]. Indeed, in a similar case with NAL-POL cycling (Supplementary Fig. 3a), the emergence of POL resistance was largely delayed, suggesting that POL resistance appeared after fitness cost of NAL resistance was compensated by secondary adaptations[41,42]. This would also suggest that, if the KAN-POL cycling was continued for a longer period, POL resistance may appear at a later stage of the experiment. The latter case of CHL-POL cycling, in which CHL resistance was completely suppressed, can be explained by the reversible nature of CHL resistance (which was also observed with POL resistance, as described below). In Supplementary Fig. 2, it was observed that developed CHL resistance was reversed back to the susceptible level when the antibiotic stress was switched to POL. This indicates that CHL resistant mutants appeared during a period of CHL stress were perished when the antibiotic was switched to POL. If the cycling rate was more frequent, it would keep the CHL resistance level low because CHL resistant mutants would be removed before they become a majority in the population.

**Mathematical model of bacterial multi-drug evolution.** From above results, we suspected that three factors were involved in the evolutionary patterns under cycled antibiotic stress, that is, (1) the duration of silent phase, (2) the rate of adaptation and (3) cross resistance and collateral sensitivity profiles. To study the obtained results from a broader perspective, we derived a simple model incorporating the three factors to describe the evolutionary patterns (Supplementary Methods). It should be noted that we employed a phenomenological model rather than commonly adopted population genetics models[8,42,43]. This is because previous studies revealed that, despite diverse underlying genetic alterations, bacterial evolution for stress resistance can exhibit remarkably similar phenotypic trajectories[7,8,44–46]. Additionally, from a prediction point of view, it would be helpful if a model is based on experimental parameters that can be measured relatively easily. Thus, we here developed a theoretical model relying only on relative MIC levels.

The model was based on two assumptions: (1) The development of bacterial resistance to an antibiotic A, termed as $R_A$, is constantly promoted during exposure to antibiotic A. However, the resistance level eventually saturates due to fitness costs[47], as we observed above. (2) Increased $R_A$ negatively affects resistance to another antibiotic B, or $R_B$ largely if the antibiotics are a collateral sensitive pair[12,13,29]. Otherwise it has small effect (neutral or cross resistant pair). Conceptually, the model can be described as a system with positive autoregulation and double negative feedback loops (Fig. 3a). The model has three key parameters, the duration of silent phase $\theta$, the rate of adaptation $\alpha$ and the collateral sensitivity/cross resistance coefficient $\beta$. These model parameters were determined based on the fitting parameters of experimental results under single antibiotic stress.

The geometric structure of the model shown in Fig. 3b,c explains the origin of the unique evolution patterns of POL resistance. The nullclines for POL resistance (yellow curve in Fig. 3b) and KAN resistance (green) intersect at three points, two of which are stable steady states (indicated as solid circles). Thus, in theory, the evolution of bacterial resistance to POL and KAN can result in either a multi-drug-resistant or single drug-resistant state. However, in the case of cycling with KAN, the bacterial population developed resistance to KAN first due to the rapid adaptation rate (determined by $\alpha$) and POL resistance

did not develop at all despite antibiotic cycling (Fig. 3d,g). In contrast, when POL was cycled with an antibiotic with slow adaptation rate, such as NIT, the evolution ends up in the multi-resistant state (Fig. 3e,h). This is because POL resistance rapidly increased before the system reached a single antibiotic state due to the slow adaptation rate of NIT. The geometric structure is unique to the case with POL which has a large silent phase (determined by $\theta$). In contrast, the nullclines have only one intersection when the antibiotics other than POL were cycled (Fig. 3c). This means that the evolution under antibiotic cycling always results in a multi-drug-resistant state in this system (Fig. 3f,i). Indeed, the cycling experiments using antibiotics other than POL always resulted in a multi-drug-resistant state so far as we tested (Supplementary Fig. 3b).

Evolutionary trajectories simulated by the model were in good agreement with experimental results in terms of the final drug-resistant states (Fig. 4) as well as the evolutionary patterns (Supplementary Fig. 6). In Fig. 4a,b, the normalized drug resistance level for one of the cycled antibiotics was plotted against that for another antibiotic. For example, KAN-POL cycling where KAN and POL are represented as drug #1 and #2, respectively, was mapped on the $x$ axis because only KAN resistance was developed while POL resistance was completely suppressed. In general, the results can be categorized into two groups, that is, the single and multi-drug-resistant states, as observed above. The former cases (for example, KAN-POL) were plotted on the axes, while the latter ones (for example, NAL-KAN) were in the upper right part of the plot. To illustrate the predictability of the theoretical model, the normalized predicted MIC values were plotted against the normalized observed MIC values (Fig. 4c,d). Overall, the model predictions showed good correlations with the experimental results ($r^2 = 0.44$ and 0.64, respectively). We also constructed a null model of the mathematical model as a comparison. In the null model, all the collateral sensitivity/cross resistance coefficient term $\beta$ were set to zero. Comparison of predictions by the null model and experimental results showed poor correlations ($r^2 = 0.03$ and 0.20. Supplementary Fig. 9). As the null model has no interaction between antibiotic resistance due to $\beta = 0$, both drug resistance levels approached to the maximum at the rate determined by $\alpha$. This result illustrates that the interaction between $\alpha$ and $\beta$ is crucial for diverse evolutionary patterns and hence demonstrates the predictability of the mathematical model.

It should be also noted that there were some cases in which the model and experiments did not match. For example, CHL-POL cycling resulted in a single drug resistance (POL resistance only) while the model predicted that the bacterial population would result in multi-drug-resistant states (Supplementary Figs 3a and 7a). As discussed in the previous section, an additional effect of bacterial resistance, such as the reversible CHL resistance, might be involved in the experimental pattern of antibiotic cycling and need to be incorporated in the model to reproduce the behaviour accurately. In the case of cycling using NAL and POL, experimental results showed that POL resistance appeared at a later stage of the experiment. However, the theoretical model predicted a single drug-resistant state (NAL resistance only). This suggests that there was some additional adaptation process that alleviated the fitness costs for NAL resistance occurred during the cycling experiment (for example, compensatory mutations[41,42]). Indeed, the model reproduced the experimental results when such effect was taken into account by reducing the collateral sensitivity/cross resistance coefficient $\beta$ to zero at a certain point during the experiment (Supplementary Fig. 6b and Supplementary Methods).

We derived an approximate analytical solution of the model, which describes that the normalized maximum resistance to

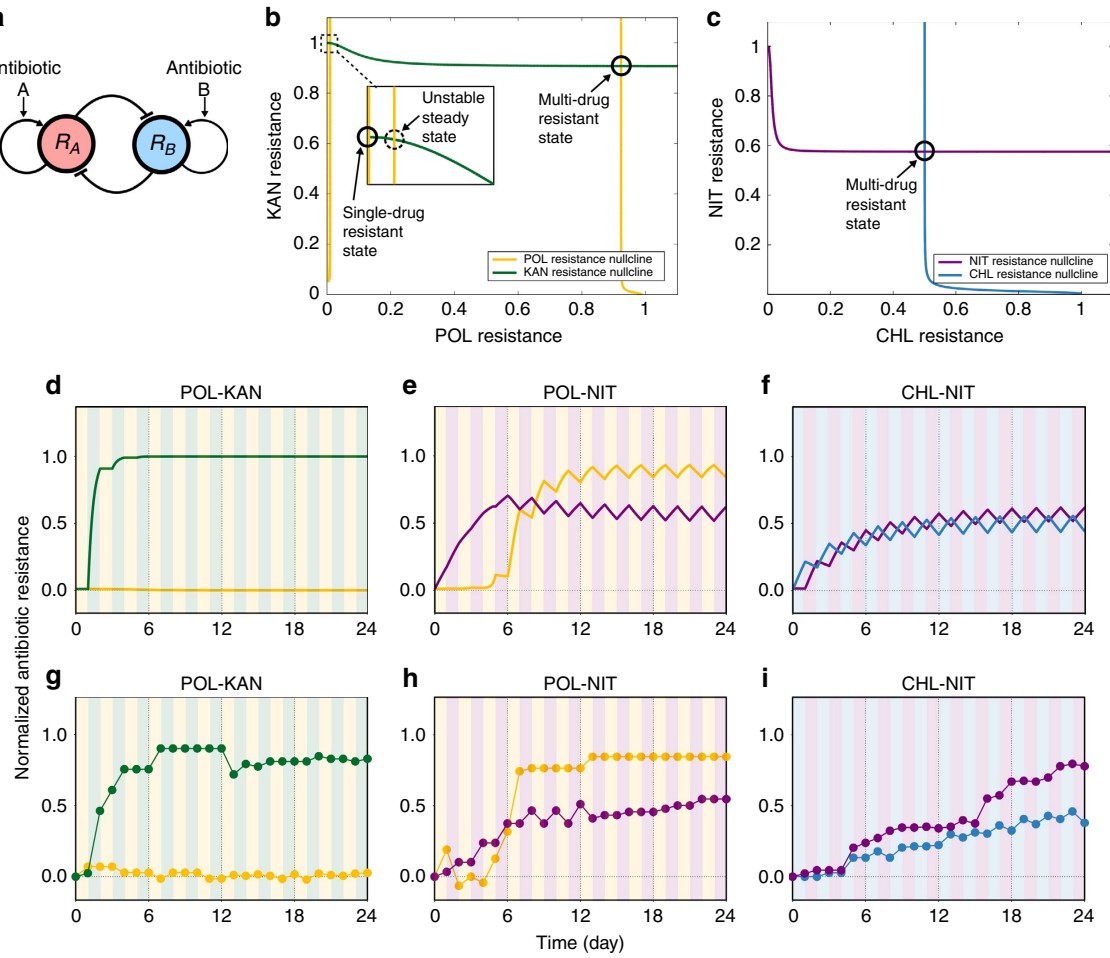

**Figure 3 | Theoretical models of multi-drug resistance evolution.** (**a**) Schematic diagram of the model consisting of positive autoregulation and double negative feedback loops. (**b**) Phase space of multi-drug evolutionary dynamics with POL and KAN. Two solid circles indicate stable steady states, corresponding to single and multi-drug-resistant states, respectively. A broken circle is an unstable steady state. MICs were normalized to one by the maximum MIC of single antibiotic evolution. (**c**) Phase space of multi-drug evolutionary dynamics with NIT and CHL. Only one stable steady state, multi-drug-resistant state, exists in this case. (**d**–**f**) Simulated evolutionary trajectories of bacterial resistance with POL and KAN, POL and NIT, and CHL and NIT, respectively. Antibiotics were cycled with 1 day interval. (**g**–**i**) Experimental data collected from the morbidostat system.

antibiotic A is determined as $R'_A = \alpha/(\alpha + \beta)$ in the simplest form (Supplementary Methods). This means that the maximum resistance level is determined by the ratio of cross resistance/collateral sensitivity profile and the rate of adaptation. Such effect of $\alpha$ and $\beta$ was indeed seen in the experimental results (Supplementary Fig. 3): Antibiotic cycling with collateral sensitive pair (that is, large $\beta$) lowered the maximum resistance level compared to that with cross resistant pair (small $\beta$), for instance, CHL and KAN, and CHL and NAL pairs, respectively. The equation also indicates that if antibiotic A with large $\alpha$ is cycled with antibiotic B with small $\alpha$, the evolution results in $R'_A > R'_B$ on the final day, which can be confirmed in the experimental results.

**Reversing drug resistance by modulating antibiotic stresses.**
One of the important implications in the model is that the evolution of bacterial drug resistance can be viewed as a dynamical system because the model proposed here is a simple deterministic system. Such dynamical systems view of biological systems was previously proposed in the context of stem cell differentiation and drug resistance evolution of caner cells[48,49]. In the current context, this view suggests that drug-resistant states of bacterial population can be directed from one state to another if

antibiotic stresses are modulated externally. In fact, it was theoretically indicated that antibiotic cycling with a pair of synergistic drugs can select susceptible bacteria while eliminating drug-resistant ones from a population[20]. To verify this experimentally, we examined the reversibility of evolved antibiotic resistance as it can be considered as a state transition between resistant states. In particular, we here mainly focused on POL resistance because the cycling with this antibiotic has two resistant states (Fig. 3b) and also because it is one of the 'last-resort' antibiotics[37]. The model predicted that an evolved resistance can be removed from a bacterial population if antibiotic cycling is switched to a single antibiotic stress (Supplementary Fig. 7a). To confirm this experimentally, we challenged bacterial populations from the cycling experiments with consecutive exposure to counteracting antibiotics. Results showed that the resistance can be indeed reversed even from the last day of antibiotic cycling at the expense of increased resistance to the counteracting antibiotic (Fig. 5a). It should be also noted that, not only single, but also multi-drug resistance in a bacterial population can be reversed by exposing to a counteracting antibiotic (POL-NIT case in Supplementary Fig. 7b). The reversibility did not depend on the evolved resistance or counteracting antibiotics used (Supplementary Figs 2b and 7b). However,

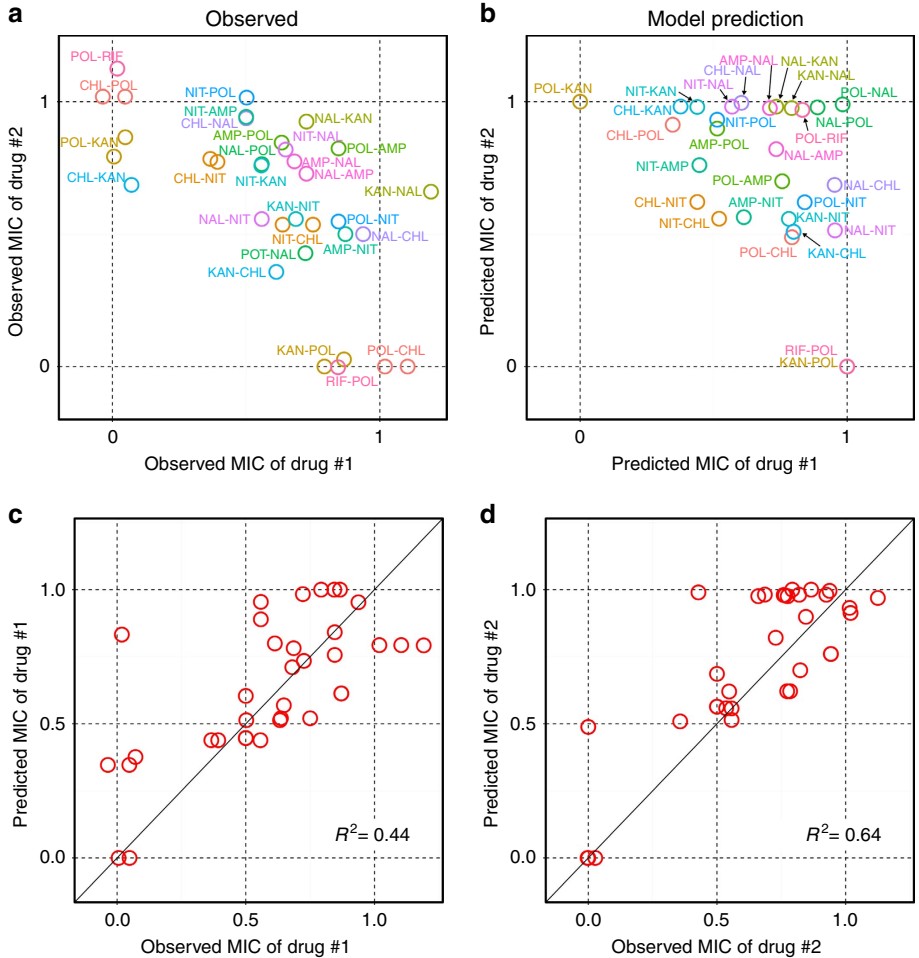

**Figure 4 | Comparison of experimental results and model predictions.** (**a**) Scatter plot of observed MICs of the first drug against that of the second drug from the final day samples of cycling experiments with 1 day interval. Note that a label for each point indicates the order of antibiotics cycled. For example, KAN-POL means that KAN was used on the first day (denoted as 'Drug #1' in the axis label) and POL was on the second day ('Drug #2'), and then cycled with the order. Due to this notation, KAN-POL and POL-KAN are plotted on the x and y axes, respectively, even though only KAN resistance was developed in both cases (Supplementary Fig. 3a). (**b**) Scatter plot of predicted MICs from the final time point of simulated results. Note that the effect of fitness cost alleviation for NAL and POL cycling was incorporated in the results. (**c,d**) Scatter plots of predicted MICs against observed MICs for the first drug and the second drug, respectively. Note that the same data used for **a** and **b** are used here.

there were also the cases when an antibiotic resistance was not reversed, especially the cases when a population was evolved under a single antibiotic stress (Supplementary Fig. 7c).

We examined the mechanism behind the evolved resistance and reversibility, and identified conditions that contributed to this phenomenon: Genetic and phenotypic heterogeneity in a bacterial population. First, we sequenced the whole-genome of POL resistant mutants evolved under single or cycling POL stress. While a number of mutations were found in the evolved strains (Supplementary Table 1), we identified mutations in *basS* and *secD* genes as key mutations that confer POL resistance (Fig. 5b and Supplementary Fig. 8). BasSR (also called PmrAB) two component system regulates the lipopolysaccharide modification pathways and is known to confer POL resistance[37]. SecD, a part of the Sec protein translocase complex, belongs to the resistance-nodulation-cell division family of multi-drug exporters[50] and is known to play an auxiliary role in antimicrobial peptide resistance[51]. The sequencing results indicated that the reversal of an antibiotic resistance was possible when there were genetic variations in *basS* or *secD* gene within a population (Fig. 5b and Supplementary Fig. 8). In contrast, the reversal was not observed in the genetically homogeneous population in terms of *basS* or

*secD* allele (Supplementary Fig. 8). This result suggests that the cycling of antibiotics prevented selective sweep and thus maintained the genetic heterogeneity in the population even after 24 day of antibiotic exposure. This result was supported by the fact that the populations evolved under a single antibiotic stress did not show the reversal of drug resistance as the population was genetically homogeneous. This genetic diversity was also reflected at the phenotype level. We observed MICs of individual isolates were diverse and differ from MIC of a whole population (black circles and a cross in Fig. 5c)[34,35]. These genetic and phenotypic diversities would partly account for the reversal of evolved resistance because the exposure to a counteracting antibiotic selects less POL resistant and more CHL resistant cells due to collateral sensitivity.

## Discussion

Although antibiotic cycling has previously been considered less effective than combination treatment of antibiotics[26,27], recent theoretical and experimental studies suggest that to the contrary, exploitation of the collateral sensitivity of various antibiotics is key to the design of effective treatment methods that can suppress

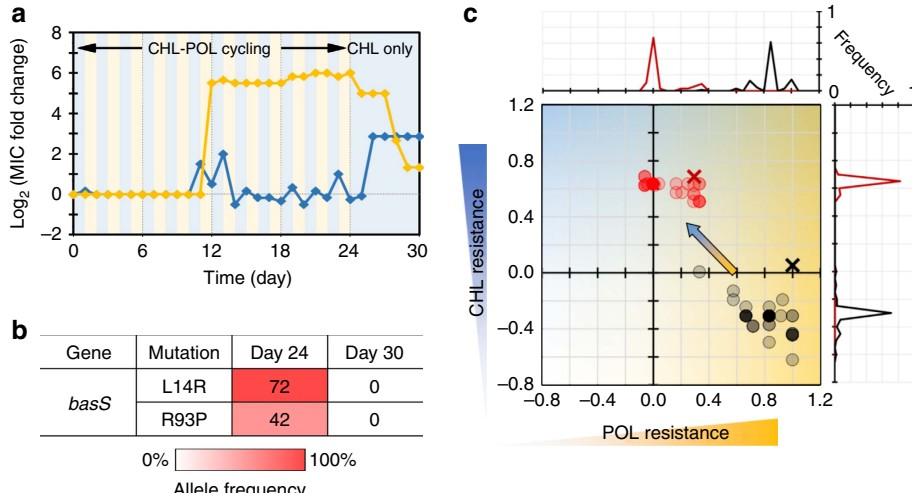

**Figure 5 | Reversal of evolved drug resistance with counteracting antibiotic.** (**a**) Evolutionary trajectories of POL and CHL resistance (yellow and blue, respectively). The background colours indicate the antibiotics used. (**b**) Allele frequencies of *basS* gene. Numbers in the map indicate the frequency of mutants in the populations, estimated from gene sequencing analysis. (**c**) Scatter plot of MICs of single isolates for day 24 and 30 (black and red circles, respectively). Samples sizes are $n = 70$ for both cases. Darker colour corresponds to higher frequency. The MIC of a whole population is shown as a black or red cross, respecitvely.

or reverse drug resistance[9,10,20]. Recent studies indicate that it is possible to revert bacterial resistance to a single antibiotic by using antibiotics in a temporally segregated manner[10,29,52]. Although collateral sensitivity has been known for more than 60 years[53], the importance of this phenomenon for suppressing the development of drug resistance was not well recognized until recently[54], which may account for failed clinical or animal studies on antibiotic cycling. In this study, we performed a series of laboratory evolution experiments for an experimental investigation into the effect of alternating antibiotic treatments. Our results show that several drug combinations can inhibit evolutionary pathways to multi-drug resistance by completely suppressing the development of bacterial resistance to one of the drugs cycled. Further, the theoretical model we developed indicated that the evolution of bacterial populations against multi-drug resistance could be not only predicted but also reversed by modulating the environmental selection stress, and, this was successfully demonstrated experimentally. The model presented here is a general model of multi-drug evolution therefore is not limited to specific drug combinations. As the model predictions of multi-drug resistance are based on evolutionary patterns under single drug stress, the model could potentially be used to predict other multi-drug evolutions where the data of single drug evolution are available. Thus, as single drug evolution experiments require fewer trials than combinatory trials, application of the model may offer a way to search for effective antibiotic combinations for cycling based on a small number of experiments.

Our findings indicate that the decreased efficacy of an antibiotic can be restored by modulating antibiotic stress. However, care must be taken when translating these results to clinical settings given that our experiments here considered *de novo* chromosomal mutations only. Further experiments, such as evolution experiments with plasmid-mediated antibiotic resistance, would be required to elucidate the feasibility of resistance suppression by antibiotic cycling in more practical situations. Evolution experiments with different modes of dosing with antibiotic cycling would be another possible route to better understand bacterial antibiotic resistance. In contrast to the morbidostat system where mutant selection is the main focus,

*in vitro* dynamic model simulates pharmacokinetic profiles of antibiotics[55], which can provide useful insights into the effect of antibiotic cycling, particularly collateral sensitive pairs, and how drug resistance may develop *in vivo*. Furthermore, antibiotic cycling experiments exploiting collateral sensitivity can also be extended to animal models. Tissue cage model[56], for example, would allow constant sampling of bacterial populations as well as drug concentration during alternating antibiotics treatments. In parallel with these possible further experiments, we foresee that deeper understanding of population dynamics under multi-drug conditions and improved theoretical modelling would potentially eliminate labour-intensive combinatorial experiments with multiple drugs. Taken together, they may lead to novel therapies that reverse bacterial multi-drug resistance.

## Methods

**Bacterial strains, culture conditions and antibiotics.** *Escherichia coli* strain MG1655 was purchased from DSMZ (Germany) and streaked to single colonies. One single colony was selected and used throughout this study. Bacterial cells were grown in 30 ml flat-bottomed glass vials (VWR, 548-0155) in a shaking incubator (Grant Instruments, ES-20) for 18 h at 30 °C before experiments in the morbidostat. Miller Lysogeny Broth (LB. $10\,g\,l^{-1}$ Tryptone, $5\,g\,l^{-1}$ Yeast Extract, $10\,g\,l^{-1}$ NaCl) was used as culture medium for all the experiments. All the antibiotics and culture media used in this study were purchased from Sigma-Aldrich unless otherwise noted. Antibiotic solutions were prepared from powder stocks. They are dissolved in solvents (Table 1), filter sterilized and kept at 4 °C in centrifuge tubes. They were periodically tested by measuring the MIC of *E. coli* (MG1655)[57]. All drug solutions were renewed before degradation.

**Construction of morbidostat system.** A morbidostat system was built based on instructions (Supplementary Fig. 1a)[58]. A morbidostat vial was constructed using a 30 ml flat-bottom with polypropylene screw cap (Supplementary Fig. 1e). The screw cap was equipped with four holes, three of which were fitted with custom-made tube adaptors consisting of needle, syringe and polytetrafluoroethylene (PTFE) sealing tape, to add or remove liquids. Two tube adaptors with short needles were used to add fresh LB medium (referred to as 'fresh medium') and fresh LB medium containing a high concentration of antibiotic ('drug medium'), respectively. A tube adaptor with long needle was used to remove excess liquid from a morbidostat vial to keep the volume of bacterial culture constant. The last fourth hole on the cap is used for filtered air intake. The entire culture tube assembly was made out of autoclavable materials.

Fresh and drug media were pumped into a vial using 12V DC peristaltic pumps (Watson Marlow, 400FD/A1) at the flow rate of $1.2\,ml\,min^{-1}$. A 16-channel

parallel peristaltic pump (Watson Marlow, 205U) was used as waste pump to remove excess liquid from morbidostat vials. Vials and pumps are connected with silicone tubing and PTFE tubing. All the pumps are computer-controlled by Arduino MEGA board (Arduino) with a custom-made mountable printed circuit board (Supplementary Fig. 1g). The custom circuit board accommodates four IC chips of eight Darlington transistor array (Texas Instruments, ULN2803A), which control up to 32 peristaltic pumps.

The optical density (OD) of bacterial cells was measured via a matched pair of an infrared LED (OSRAM, SFH4550) and a phototransistor (Vishay Semiconductor, BPW96B). Tube holder assemblies that accommodates infrared light source and photodiode were made by three-dimensional printer (Stratasys, Connex 500) using VeroBlack material (Supplementary Fig. 1f). The infrared light source and detector were positioned at 135° angle to maximize the detector sensitivity. The light-induced voltage changes on the detector are measured from analogue inputs of another Arduino MEGA board. The illumination and detection circuits are shown in Supplementary Fig. 1h. The voltage values are later converted to OD based on a calibration curve (Supplementary Fig. 1i).

The final assembled apparatus accommodates 16 morbidostat vials in a temperature controlled incubator (Lucky Reptile, HN-2UK, Supplementary Fig. 1c,d). Cell cultures in the morbidostat vials are continuously stirred by small stir bars during operation. Magnetic stirrers installed directly under the morbidostat vials are made with computer fans (SUNON, KD0504PFS2.11.GN), three-dimensional printed magnet holders and neodymium magnets (COMUS, M1219-4).

**Operation of morbidostat system.** Glass vials, tubing and bottles used for experiments were autoclaved before use. After autoclaving, silicone and PTFE tubing for morbidostat were sterilized with Virkon (Fisher Scientific) and ethanol individually, then rinsed with sterile water twice. Bacterial cells stored at $-80$ °C were thawed and transferred into autoclaved morbidostat vials with 12 ml of LB medium, in which bacterial cells were diluted 60-fold. The assembled vials containing cell cultures were then placed into the morbidostat vial holders.

A custom software that monitors bacterial growth in the vials and controls pumps via Arduino boards was written in Python. This software implements the morbidostat algorithm as specified (Supplementary Fig. 1b)[50], which was operated as follows: the turbidity of each cell culture was continuously measured every 2 s, then averaged over 30 samples (that is, average over 1 min). Every 12 min, bacterial culture in a vial was diluted by 1 ml of either fresh or drug media. No medium was introduced if the OD of cell culture is below 0.15 to avoid washing out bacterial cells from the culture. At the time of dilution, two conditions were checked: (1) If the current averaged OD is above a threshold OD ($OD_{THR} \sim 0.4$), and (2) if the current OD is larger than the OD at the time of previous dilution. If both conditions are met, 1 ml of drug medium is added to the cell culture. Otherwise fresh medium is added. Note that we used a higher OD threshold than the previous study ($OD_{THR} = 0.15$)[7]. This is because some antibiotics show abrupt changes in the dose response curve (for example, ampicillin and KAN, Supplementary Fig. 13), which makes the control of growth inhibition difficult in morbidostat if a $OD_{THR}$ is low.

The concentration of antibiotic in the drug medium was started from 10 times of the MIC at the beginning of the long-term experiment. If the bacterial population gains antibiotic resistance and the drug concentration inside the morbidostat vial is more than 60% of the concentration of drug medium, then the antibiotic concentration in the drug media was increased to twofold of that of the previous concentration (Supplementary Fig. 14).

A single morbidostat experiment was run for 22 h. No apparent formation of biofilms on the inner wall of morbidostat vial was observed at the end of experiment. Cell cultures were sampled from all the 16 vials in the morbidostat and stored as glycerol stocks at $-80$ °C.

**Morbidostat data analysis.** Morbidostat operation log data were analysed by custom Matlab scripts. The growth rate $\mu$ (unit: h$^{-1}$) was calculated as $\mu = \frac{\log(OD_n^E / OD_n^S)}{\Delta t} \times 3600$ where $OD_n^E$ is an OD value of cell culture at the $n$-th dilution and $OD_n^S$ is that of the same cell culture 10 min before the $n$-th dilution. $\Delta t = 600$ s. The concentration of an antibiotic A at $n$-th dilution $[A]_n$ was calculated by the following equation.

$$[A]_n = \begin{cases} \left( V[A]_{n-1} + [A]^{stock} \right) / (V + \Delta V) & \text{if drug media is added} \\ V[A]_{n-1} / (V + \Delta V) & \text{if fresh media is added} \end{cases}$$

Where $V$ is the volume of cell culture in the morbidostat vial (that is, 12 ml), $\Delta V$ the volume of fresh or drug media added at each dilution (1 ml), $[A]^{stock}$ is the concentration of the antibiotic A in the stock drug medium (varies). Statistical tests, that is, s.e.m. and $P$-value by Welch's two sample $t$-test, were performed using R.

**MIC measurement.** The MIC of samples collected during evolution experiments was measured by growing the samples on a set of LB agar plates with varying antibiotic concentrations ($\sqrt{2}$-fold serial dilution). The evolved strains from the morbidostat experiments stored at $-80$ °C were thawed and transferred to 384-well microplates (Fisher) with 65 μl of LB medium. The dilution rate here

was 1:50. The microplates were incubated at 30 °C for 18 h with shaking at 1,000 r.p.m. before the MIC agar plate assay. LB agar plates were prepared in omni plates (Fisher). The LB agar medium were autoclaved and antibiotics were added after the agar was cooled down. Cells grown in the 384-well plates were transferred to agar plates using the pin replicator (V&P Scientific)[59]. The agar plates were incubated at 30 °C for 18 h. After the incubation, cell growth was evaluated by presence or absence of visible colonies. The minimum antibiotic concentration without a visible colony was determined as the MIC. Four replicated experiments were performed for each measurement and averaged values were calculated. The antibiotic resistance of an evolved mutant was calculated as log$_2$-transformed fold changes in MIC relative to wild-type strains, that is, $\log_2 \left( \frac{MIC[A]_{EVO}}{MIC[A]_{WT}} \right)$ where $MIC[A]_{WT}$ and re $MIC[A]_{EVO}$ indicate the MICs of wild-type (ancestral) and evolved mutant against antibiotic A, respectively.

When comparing experimental results with theoretical model simulations, both data were normalized to one. For experimental data, relative antibiotic resistance values were normalized by dividing a MIC value with the maximum MIC value of averaged data from single antibiotic stress evolution.

**Logistic fitting of experimental data.** The evolutionary trajectories of bacterial resistance to an antibiotic was fitted with a logistic function where $L$ indicates the maximum relative MIC level, $k$ the steepness of the curve (Hill slope), and $m$ the midpoint of the curve. Fitting was performed using a custom-made Python script using SciPy package. The duration of the silent phase $\theta$ was calculated as the time point when $f(x) \geq 0.1$, that is, $\theta = m - \frac{\log(10L - 1)}{k}$.

**Single clone profiling.** The evolved strains from the morbidostat experiments stored at $-80$ °C were streaked on LB agar plates to pick individual colonies for MIC assay. After 15 h of incubation at 30 °C, at least 25 colonies from each sample were chosen at random and dissolved into 180 μl of LB in 96-well plate individually. After overnight incubation of the 96-well plate at 30 °C, we remove 65 μl of those cell samples into empty 384-well plate for MIC assay as described in 2.5.

**Genome extraction and whole-genome resequencing.** Total 40 samples were sequenced using Illumina MiSeq desktop sequencer: The ancestor, day 12 samples of all 19 populations of single antibiotic conditions, and day 24 samples of 20 populations of antibiotic cycling conditions and control. Each evolved strain was diluted 1:100 in 5 ml of LB with no drug and grown overnight (18 h) at 30 °C before DNA extraction. We confirmed that the overnight growth without antibiotics does not change the antibiotic resistance level by two methods. First, the resistant samples evolved under single antibiotic conditions (Supplementary Fig. 5, the final day samples) were grown overnight with and without antibiotics. MIC values were the same regardless of antibiotic stress (Supplementary Fig. 10). Second, the colony forming unit of a resistant strain was measured to see if the resistance level changes during overnight incubation without antibiotics. The final day sample of POL and CHL cycling (Fig. 5a, day 24) was used and the frequency of survivors was compared between a frozen stock and overnight culture of the stock grown without antibiotics. However, no significant difference was observed between the two conditions (Supplementary Fig. 11). Genome DNA was extracted from 1-ml cultures using Wizard Genomic Purification Kit (Promega #9FB022). RNase digestion step was added after the genome extraction process. Sequencing libraries were prepared using Nextera XT DNA Library Preparation Kit.

**Bioinformatic analysis of genome sequencing.** The obtained sequences were aligned to the MG1655 reference data (NC_000913.2). Alignment was performed using Bowtie2. SNPs and indels were identified using SAMtools. Identification of mutations were analysed using breseq (ref. 60). A minimum coverage of 100-fold was accomplished with each strain. Selected putative variants (SNPs and indels) detected by whole-genome resequencing were verified by PCR followed by Sanger sequencing. Primers used for Sanger sequencing were designed using Primer3web.

**Determination of allele frequency.** Allele frequencies were determined from fluorescent signal intensities of Sanger sequencing data, that is, the ratio of a signal intensity of a mutant SNP and total signal. Sequences were aligned by DNA Baser (www.dnabaser.com/). Although this is not an accurate way to determine allele frequencies, the analysis was used to determine whether an evolved population is genetically homogeneous or heterogeneous.

**Mathematical model.** Numerical solutions for the theoretical model were calculated in Matlab (Mathworks, USA) and in Julia (http://julialang.org/) using Sundials package. Further details of the model can be found in Supplementary Methods.

**Data availability.** Whole-genome sequencing data have been deposited in the National Center for Biotechnology Information (NCBI) Sequence Read Archive (SRA) with the following sample accession codes: SRS1749080 (CHL), SRS1749079 (POL), SRS1749078 (POL-CHL), SRS1749077 (CHL-POL_D6), SRS1749084

(CHL-POL_D10), SRS1749083 (CHL-POL_D18), SRS1749082 (CHL-POL_D24), SRS1749081 (MG1655 wild type) and SRS1749076 (Control). Custom software used in this study is available from http://www.chem.gla.ac.uk/cronin/e_media/. The authors declare that all other relevant data supporting the findings of the study are included in this published article and its Supplementary Information, or are available from the corresponding authors upon request.

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

## Acknowledgements

The authors gratefully acknowledge financial support from the EPSRC (Grant No's EP/H024107/1, EP/I033459/1, EP/J00135X/1, EP/J015156/1, EP/K021966/1, EP/K023004/1, EP/K038885/1, EP/L015668/1 and EP/L023652/1), BBSRC (Grant No. BB/M011267/1), the EC (projects 610730 EVOPROG, 611640 EVOBLISS), ERC (project 670467 SMART-POM), University of Glasgow for LKAS fellowship, the Honjo Scholarship Foundation and JSPS Grant-in-Aid for Scientific Research (B) [26290071]. Authors thank Salah Sharabi for designing Arduino shield, Hazuki Kotani for assistance with genome sequencing, Kazufumi Hosoda and Stephanie Connelly for valuable comments and discussions.

## Author contributions

L.C. conceived the concept and L.C. and S.T. designed the experiments. M.Y. and S.T built the morbidostat system. M.Y. and S.G.R. performed experiments and subsequent assays. M.Y. and S.T. analysed the experimental data. L.C. and S.T. conceived the theoretical model approach and S.T. developed and analysed the model. M.Y., S.G.R., T.H. and C.F. performed genome sequencing analysis. All the authors prepared and commented on the manuscript.

## Additional information

**Competing interests:** The authors declare no competing financial interests.

