## [Peer review file · Nature Communications]

Reviewers' comments:

Reviewer #1 (Remarks to the Author):

GENERAL COMMENTS

A main concern is that the field may have moved beyond static modeling, since the emergence of resistance is commonly studied using in vitro pharmacodynamic models (see papers from the laboratories of Firsov, Tam, and Drusano). The present work needs to be placed within the context of fluctuating drug concentrations, which are being used to study combination therapies (see work by Firsov). It may be possible to reframe the work as relevant to the limited application of continuous infusion.

The Introduction needs to provide a better historical context, since some readers will believe, from clinical experiments, that antibiotic cycling does not work and has been abandoned. That point of view needs to be overcome to establish a readership. It may be possible to reframe the work more toward bacterial evolution using antibiotics as tools, since that would reduce the need for clinical relevance.

The possibility of cross-resistance needs to be considered more carefully. Even though the main targets of the compounds may differ, efflux mechanisms may not discriminate as well. Once the first mutation is acquired, for example efflux, the probability of acquiring subsequent target alleles increases substantially. This is an issue that derives from the "hill-climbing to resistance" concept coupled with the mutant selection window.

The manuscript needs to be prepared more carefully to establish author/journal credibility. In particular, English usage is a little bit off, especially in the use of articles. Hyphenation and comma errors should be corrected. The use of acronyms, often undefined, should be avoided. They almost always require unnecessary thinking and retard the understanding of the work. I would also recommend the use of continuous line numbering to facilitate review. Below are specific comments that may be useful during revision.

SPECIFIC COMMENTS

Page 1. Interim? This seems like a strange term, since combination therapy has been used for decades with tuberculosis, and there seems to be no end in sight. Moreover, the general high-level global consumption is unlikely to drop. Thus, the word interim undermines author credibility.

Page 5 middle. Use of the word "this" is vague.

Discussion paragraph 1: threat. Authors are vague here and perhaps miss the point. Drug toxicity to patients is an over-riding consideration. When a pathogen is multidrug resistant, the physician must go to agents that are likely to have greater toxicity for the patient.

Discussion new bacterial resistance. This problem needs to be solved for this work to be of general utility.

Methods: scientific names should be italic

Methods: give a reference for lysogeny broth composition

Methods: *E. coli* should have a space and be in italic

Operation ... growths?

MIC measurements: define more carefully what you mean by gradient of concentrations. Do

you have gradients in the plates? If so, how do your MIC values relate to standard measurements?

Visible colonies – with the pin applicator do you see individual colonies or are you simply scoring a mass of growth? If the latter, is there concern about the effect of inoculum size? If you actually see colonies, how many were there at sub-MIC concentrations?

Bottom of page 15. Use of the term resistance will be seen by some readers as careless. This is because resistance is defined clinically as isolates whose MIC exceeds an empirical breakpoint. An isolate is either resistant or it is not. The term susceptibility (or decrease in susceptibility) is used to be clear. Biochemists tend to ignore this distinction.

Mutant frequency – do you mean mutation frequency? If this method is not accurate, as the authors state, then the section should be revised to stress what you really measured. The current phrasing is not professional.

Reviewer #2 (Remarks to the Author):

In this manuscript the authors combine experimental evolution and theoretical modelling to study how bacterial populations adapt to fluctuating environments. In particular, this study focuses on evaluating the efficacy of sequential treatments to control resistance adaptation, a strategy that alternates between different antibiotics with the aim of reducing the overall fitness of the population by producing evolutionary mismatches. The main result of this study is that certain combinations of antibiotics can reverse multidrug resistance. This is a potentially interesting observation that could, in principle, help in the design of rational drug usage strategies to fight clinically relevant pathogens. However, this observation has been previously reported using different bacteria and drug combinations and therefore, although methodologically sound, I believe this study is not conceptually novel. Indeed, in the last couple of years a series of published papers have demonstrated, both theoretically and experimentally, that drug resistance can be controlled by appropriately cycling between different antibiotics, even in stringent scenarios where the drugs used are structurally similar (10.1371/journal.pone.0122283, 10.1371/journal.pcbi.1004493) or select for cross resistance (10.1371/journal.pbio.1002104). None of these papers has been cited in this manuscript.

The continuous culture device constructed by the authors is very interesting, as it allows implementation of feedback controllers that dynamically manipulate the environmental conditions from real-time estimates of bacterial density. Previous studies have focused on implementing serial dilution protocols with daily transfers, and therefore imposing a temporal structure characterised by growth-dilution cycles, but also limiting the switching rate to periods longer than one day. Surprisingly, despite the possibilities offered by their experimental system, the authors choose to implement periodic cycles with daily switches because “shorter intervals tended to delay the development of the antibiotic resistance” (page 5). This is a strange reason to ignore rapid switching rates, in particular given that the main goal of the study is “to suppress the development of drug resistant mutants” (page 2).

The mathematical model posed by the authors is based on positive autoregulation and double negative feedback loops that seems to be in agreement with the experimental data (Figure 2). There are some cases where the data and the model produce different results (which is expected in a simple, phenomenological model) and the possible sources of disagreement are discussed in the Supplementary Material (Supp. Fig. 6a and Fig 3a). I would suggest this discussion should be included in the main text. An important assumption of the theoretical model is the existence of a fitness cost associated with resistance, considered by imposing the constraint that “the resistance level eventually saturates” (page 6). I understand that this is a parsimonious modelling assumption, although (as the authors mention) some drug resistance mutations are essentially cost-free and, even if they are costly, the cost can be readily compensated. In any case, an immediate consequence of this assumption is that a multidrug resistant subpopulation would present a lower fitness than single-drug resistant strain in single-drug environments, and therefore it's not surprising that multidrug resistance can be reversed when only one antibiotic is deployed. Actually, it has already been shown mathematically that in a 2-locus, 2-allele model of a continuous culture device where drug-resistance is costly, that a multidrug resistant subpopulation can be minimised by appropriately cycling antibiotics, even when the drugs used interact synergistically (10.1098/rsif.2012.0279).

I have serious concerns about the sample preparation used for the single-clone profiling and whole-genome sequencing. The authors describe in their methods (page 16) that evolved strains were diluted in LB with no drug and grown overnight before DNA extraction and performing MIC assays. I understand the need for greater densities in order to obtain large DNA quantities necessary for Illumina MiSeq sequencing, but removing the selective pressure in favour of resistant mutants could have significant consequences on the MIC of the evolved population, as well as in the mutations identified from genome sequencing. This step is critical in particular when, as argued by the authors, drug-resistance mutations are associated with a fitness cost (see, for example, [10.1093/gbe/evu106](https://doi.org/10.1093/gbe/evu106)).

My main concern about this manuscript, however, is that the authors ignore a considerable body of literature relevant to the antibiotic cycling problem and yet they state in the introduction that “systematic investigations into how a bacterial population develops resistance to multiple cycling antibiotics have yet to be done”. Indeed, this is a very difficult problem and one that needs more attention from scientific and medical communities and, although I think this study could be a contribution in the search for rational antibiotic deployment strategies, I also believe it is imperative that the authors put their findings in the context of previously published studies.

Reviewer #3 (Remarks to the Author):

Yoshida and coworkers performed an extensive study in which they experimentally and theoretically elucidated how different temporal antibiotic dosing schemes affect the development of antibiotic resistance in *E. coli*. Motivated by recent experimental studies, they constructed a continuous-culture device, which enabled them to dynamically adjust drug concentrations both on short time scales (to ensure constant selection pressure) and on longer time scales (to switch between different antibiotics). This custom-built device that is similar to the “morbidostat” (Toprak et al., *Nature Genetics*, 2012) provides experimental flexibility and is well suited for assessing how the duration of antibiotic treatment prevents or promotes resistance evolution to sequential antibiotic treatments. The authors demonstrated that temporal cycling of drugs promotes genotypic and phenotypic heterogeneity, which is at the heart of resistance reversal. Their findings are supported by genotyping of bacterial populations and by a phenomenological model of resistance evolution in a multidrug environment.

The experimental part of the study appears largely well done with clearly described methods (where these deviate from reported ones). Careful experimental and theoretical studies are needed to understand and combat antibiotic resistance evolution; hence, this study will likely be interesting for a broad readership. The generated dataset is quite comprehensive and will represent a good reference for further studies of antibiotic resistance evolution.

However, I have several concerns (see below) that need to be clarified to corroborate the proposed claims and increase confidence in the reliability of the data.

Major points:

1. The result that absolutely no resistance to one of the drugs evolves e.g. when KAN and POL or CHL and POL are switched would be quite striking but needs additional clarification. For one, conceptually similar experiments (using different drugs) were reported in the cited reference (Kim et al., *PNAS*, 2014) but this study consistently found increases in resistance to both drugs (as one would generally expect). What causes this difference? Some plausible explanation for this would be important. Also, the data in Fig. 1c,e that show these results have features that require some explanation: in panel c, POL resistance increases by ~2-fold on day 10 but then decreases again on day 11 even though POL is present at that time. It would help to explain how that can happen because this selection against these first resistant mutants in the presence of the drug may be important for the observed phenomenon. In panel e, CHL resistance seems to start increasing right for the last data point on day 24. Overall, it would greatly help to explain the plausibility of these results in the framework of basic population genetics (see below).

2. The theoretical model described in the Supplement contains parameters that are closely related to the parameters of the logistic equation used to quantify the abruptness of the antibiotic resistance onset. Are there any analytical ways to connect the parameters L , k and m of the logistic equation with the parameters n , θ and α from the dynamical system? It would help to reduce the number of parameters/equations to keep the text more concise and easier to follow.

In the model, the Hill coefficients are first introduced as free parameters, but are later set to a constant value for all cases. The same applies for γ , k_A and k_B ; how were these values determined? A quantitative assessment of the impact of parameter values should be

provided.

Collateral sensitivity enters into the model through the coupling constant β , which is given as $\beta = \gamma 2^{-x}$, where x is a value from the collateral sensitivity/resistance matrix in Fig. 6. Values in the matrix are \log_2 transformed MIC-fold changes; hence, β directly equals γ times the MIC fold change. However, the reported values for CHL-POL and POL-CHL do not agree with this (given that $\gamma = 0.1$).

Setting all coupling constants β to 0 might be a useful null-model when comparing correlation between the observed and predicted MIC of drugs (Fig. 3). Such a null model would be useful to assess how successful the theoretical predictions are.

It would help to emphasize that this model is phenomenological and it should be explained why the authors chose this model over well-established population genetics models for such evolution experiments. A population genetics model could provide deeper insight into the evolutionary dynamics, in particular the peculiar dynamics observed for POL (Fig. 1e): e.g. what is the beneficial mutation rate that would lead to such dynamics? The effect that POL resistance evolves perfectly reproducibly after exactly 7 days in this experiment is potentially interesting but deserves some explanation: is this plausible given the inherent stochasticity of evolutionary dynamics?

3. In the Discussion section, the authors argue: "From a clinical point of view, our finding is crucial because it suggests that the decreased efficacy of an antibiotic can be restored, even from a multi-drug resistant bacterial population, by adopting a time-modulated drug dosing strategy." This claim may be true for de novo developed resistance in mixed populations and less so for clinically troublesome multi-drug resistant bacteria. Already the text on page 11 of the manuscript indicates that reversibility is possible but in a very special regime of a mixed population (which is maintained due to antibiotic cycling). These claims of clinical relevance should be supported by corresponding data or toned down.

4. In general, when stating and comparing quantities (e.g., k rates) errors and, where applicable, appropriate statistical tests of the claims should be reported. In order to maintain the flow of the text, the statistical tests used and the p -values could be reported in the supplement.

5. The cultures used in genome extractions were grown overnight in the absence of antibiotic. Was it verified that resistance occurrence in the population was not lowered (if associated with the fitness cost), e.g., by comparing the ratios of colony forming units on plates with and without antibiotic before and after overnight incubations?

6. In the morbidostat protocol that is used, what is the growth rate in the absence of the antibiotic? What is the relative growth inhibition given the dilution and antibiotic concentration adjustment protocol described in the Method section? Is the relative inhibition indeed identical for all used antibiotics? What is the average number of antibiotic concentration adjustments needed to achieve constant inhibition? Is this time comparable with the switching rate?

7. Some of the chosen antibiotics (kanamycin, ampicillin) are known to have a very steep dose-response curve, i.e. the growth rate as a function of antibiotic concentration has a very abrupt drop, which in turn makes it hard to control growth inhibition in a device like the morbidostat. How was this problem solved? Sample growth curves for the single antibiotic

regime for all used antibiotics should be provided (i.e. Supp. Fig. 1b with real data).

8. Figure 1 suggests that the MIC of CHL and NIT increased roughly 16 fold over the course of 12 days. Data from the cited references (Toprak et al., Nature Genetics, 2012) and (Chevereau et al., PLoS Biology, 2015) shows that the IC50 increased by a considerably larger factor for both drugs in similar experiments. It would be helpful to explain the (potential) origin of these discrepancies. Supplementary Figure 5 states that the original data is shown in Fig. 1, which is not the case for RIF, AMP and TET. It would be important to provide the absolute increases in MIC for all drugs for cross-validation with existing literature.

9. Collateral sensitivity/resistance profiles were measured before (e.g. in the cited reference Imamovic and Sommer, Sci. Trans. Med. 2013 and in Oz et al., Mol Biol Evol, 2014). It would be helpful to provide a scatter-plot comparing the measured values with the reference values. These previous studies established that resistance to aminoglycosides (e.g. KAN) renders strains very sensitive to many antibiotics of other classes; this is less clear in the data presented here and it would be helpful to comment on this.

Minor points:

1. In the second paragraph of the introduction, it is worth mentioning that keeping bacterial cultures in the exponentially growing regime is crucial to achieve a well-controlled physiological state.

2. If x and m in the logistic equation are given in the units of time (e.g., days) then values for k should be reported with units of inverse time which is not the case. If some normalization was used that renders values unit-less, this should be reported.

3. In the section "Evolution against single antibiotic ..." evolutionary patterns are divided into two categories according to the rapidness and length of lag phase: were the k and m values used as a classifier? If so, the threshold values should be reported.

4. Comparison of the experimental results in Fig. 3 should be done more quantitatively by reporting the corresponding correlation coefficients or other metrics.

Also, figure 3 should be explained in a bit more detail; including a concise explanation why pairs of cycled drugs appear twice in inverse order would facilitate the understanding of this important figure. Also, why are these pairs sometimes very similar (e.g. NIT-CHL, CHL-NIT) and sometimes extremely different in these plots (KAN-POL, POL-KAN)?

5. Morbidostat data analysis: Is the expression for the growth rate missing a logarithmic transformation of OD values? If the written equation was actually used, please report the regime in which such linear approximation is valid.

6. Supplementary table 1: Chloramphenicol acts on the 50S ribosomal subunit, more precisely on peptidyl transferase center, not on 30S ribosomal subunit as reported in this table.

7. There are some language errors (a few examples are listed below) that should be

corrected throughout:

- a. Supp.Fig.6: Blue and red grids indicate that the antibiotic pair is collateral[ly] sensitive and cross resistance [resistant?], respectively.
- b. Supp.text, p 17: "absent (i.e. = 0), the resistance"
- c. Main text p 14: "After autoclave [autoclaving?], silicone and PTF"
- d. Main text p 16 "96 well plate at at 30 °C, we remove[d]"

Responses to the reviewer comments:

We would like to thank all the reviewers for critical comments for improving our manuscript. We have carefully considered the comments and compiled our responses below. The manuscript was revised to incorporate the suggested comments. We also performed additional experiments suggested by the reviewers, which data are added to the revised manuscript. We hope the reviewers agree we have addressed all their comments and are happy with our improved manuscript.

Reviewer #1:

GENERAL COMMENTS

A main concern is that the field may have moved beyond static modeling, since the emergence of resistance is commonly studied using in vitro pharmacodynamic models (see papers from the laboratories of Firsov, Tam, and Drusano).

Our continuous cell culture system is not designed to characterise pharmacokinetics /pharmacodynamics (PK/PD), but to study long-term evolutionary dynamics of bacterial adaptation to external stresses (in this case, antibiotics) from experimental evolution point of view. Continuous cell culture systems have been widely used in the field of bacterial experimental evolution, not just for antibiotic resistance stresses but also for other types of conditions, because it provides simple and constant environment (cf. 10.1016/j.ygeno.2014.09.015). We apologise as we think there is a misunderstanding here related to the way we framed this in the previous submission.

The present work needs to be placed within the context of fluctuating drug concentrations, which are being used to study combination therapies (see work by Firsov). It may be possible to reframe the work as relevant to the limited application of continuous infusion.

We were aware of this work, but did not refer to this in our first manuscript because those studies were mainly focusing on relatively short-term dynamics of bacterial population exposed to antibiotics. However to put our research in a broader context, three papers by Firsov et al., three review papers on the PK/PD model by Gloede et al. and Drusano et al. are now cited as in the introduction to provide a historical context (see also our next response below).

The Introduction needs to provide a better historical context, since some readers will believe, from clinical experiments, that antibiotic cycling does not work and has been abandoned. That point of view needs to be overcome to establish a readership. It may be possible to reframe the work more toward bacterial evolution using antibiotics as tools, since that would reduce the need for clinical relevance.

We reframed the introduction by emphasising that our main motivation for this study was bacterial evolution. (Line 48-49, "This study aims to address this, and to gain fundamental insights into bacterial evolution under fluctuating antibiotic stress, by the design of a series of long-term laboratory evolution experiments. ")

Additionally, another motivation was to systematically investigate the effectiveness of temporally-modulated antibiotic stresses to suppress the emergence of bacterial drug resistance. Indeed, some clinical tests and mathematical models concluded antibiotic cycling does not work. However, to our best knowledge, there are still on-going discussions about its effectiveness because of conflicting results between theoretical models (cf. 10.1371/journal.ppat.1004225) as well as between experimental tests (10.1086/504484). We speculated that it would be partly due to inappropriate choice of

drug combinations. Recently, a systematic investigation on cross resistance and collateral sensitivity for antibiotic cycling was conducted (10.1126/scitranslmed.3006609), but the long-term effects of antibiotic cycling based on these combinations have yet to be tested. Hence, we performed the evolution experiments to obtain fundamental insights into bacterial evolution under antibiotic cycling stresses.

In summary, we have made changes to the introduction to reflect above points as follows:

1. Added history of antibiotic resistance studies
 - a. PK/PD model study (Firsov et al.)
 - b. Mathematical modelling (Drusano et al.)
 - c. Bacterial evolution study
2. Emphasis on experimental evolution as the motivation of this study

The possibility of cross-resistance needs to be considered more carefully. Even though the main targets of the compounds may differ, efflux mechanisms may not discriminate as well. Once the first mutation is acquired, for example efflux, the probability of acquiring subsequent target alleles increases substantially. This is an issue that derives from the “hill-climbing to resistance” concept coupled with the mutant selection window.

We indeed paid special attention to the possibility of cross resistance. For instance please refer to Supplementary Fig.6, the cross resistance and collateral sensitivity profile. For example, tetracycline(TET)-resistant bacteria (top row) showed cross resistance to many other antibiotics (coloured as red), and most of drug resistant bacteria showed cross resistance to rifampicin (RIF, second rightmost column). For these reasons, we did not conduct any combinatorial experiments with TET/RIF except RIF-POL combination.

The manuscript needs to be prepared more carefully to establish author/journal credibility. In particular, English usage is a little bit off, especially in the use of articles. Hyphenation and comma errors should be corrected. The use of acronyms, often undefined, should be avoided. They almost always require unnecessary thinking and retard the understanding of the work. I would also recommend the use of continuous line numbering to facilitate review. Below are specific comments that may be useful during revision.

We apologise for the typos and grammar errors. We have carefully checked the manuscript again and corrected these problems. Line numbers were added in the manuscript and supplementary text to aid further review.

SPECIFIC COMMENTS

Page 1. Interim? This seems like a strange term, since combination therapy has been used for decades with tuberculosis, and there seems to be no end in sight. Moreover, the general high-level global consumption is unlikely to drop. Thus, the word interim undermines author credibility.

Corrected as follows (Line 24-25):

The use of a combination of drugs as a ‘multidrug strategy’, such as combination therapy and antibiotic cycling, has been proposed and used to cope with the current situation

Page 5 middle. Use of the word “this” is vague.

Corrected as “the single drug resistant state with POL”.

Discussion paragraph 1: threat. Authors are vague here and perhaps miss the point. Drug toxicity to patients is an over-riding consideration. When a pathogen is multidrug resistant, the physician must go to agents that are likely to have greater toxicity for the patient.

We removed the sentence including the term and also reduced the reference to clinical cases to emphasise the bacterial evolution aspect of our study.

Discussion new bacterial resistance. This problem needs to be solved for this work to be of general utility.

We agree with the reviewer. Our result may suggest a key to suppress/delay/reverse drug resistance, but we also think we should be careful when translating our results to practical situations and further experiments are still required for our work to be of general utility. Thus, we added following sentence in the Discussion section:

Line 370-374:

Our findings indicate that the decreased efficacy of an antibiotic can be restored by modulating antibiotic stress however care must be taken when translating these results to clinical settings given that our experiments here considered de novo chromosomal mutations only. Further experiments, such as evolution experiments with plasmid-mediated antibiotic resistance, would be required to elucidate the feasibility of resistance suppression by antibiotic cycling in more practical situations.

Methods: scientific names should be italic

Methods: give a reference for lysogeny broth composition

Methods: E. coli should have a space and be in italic

Operation ... growths?

Corrected.

MIC measurements: define more carefully what you mean by gradient of concentrations. Do you have gradients in the plates? If so, how do your MIC values relate to standard measurements?

We are sorry for the confusing description of our method. This method is similar to the broth micro-dilution method, but uses a set of agar plates with varying antibiotic concentrations ($\sqrt{2}$ -fold serial dilution). We modified the sentence as follows:

Line 456-457:

The MIC of samples collected during evolution experiments was measured by growing the samples on a set of LB agar plates with varying antibiotic concentrations.

Visible colonies – with the pin applicator do you see individual colonies or are you simply scoring a mass of growth? If the latter, is there concern about the effect of inoculum size? If you actually see colonies, how many were there at sub-MIC concentrations?

We saw individual colonies after overnight incubation. With our 384 pin replicator, approximately 0.1 ul of cell culture was applied on the plate. If bacteria did not grow on a spot, the culture was dried out and there was no trace of bacteria on the spot. MIC concentration was defined as the lowest antibiotic concentration that bacterial growth did not occur. Below is a picture of the agar MIC measurement for reference.

Bottom of page 15. Use of the term resistance will be seen by some readers as careless. This is because resistance is defined clinically as isolates whose MIC exceeds an empirical breakpoint. An isolate is either resistant or it is not. The term susceptibility (or decrease in susceptibility) is used to be clear. Biochemists tend to ignore this distinction.

We used the terms based on the definition commonly used in the field of bacterial evolution, in which a wild type strain was defined as a susceptible strain. We added our definition in the "MIC measurement" of Method section to clarify this.

Line 454-455:

First, we defined a wild-type MG1655 strain as a susceptible strain. Resistant strains were defined as any (evolved) bacterial samples whose MIC exceeds that of the wild-type strain.

Mutant frequency – do you mean mutation frequency? If this method is not accurate, as the authors state, then the section should be revised to stress what you really measured. The current phrasing is not professional.

We apologise for the confusing term. We changed the term as "allele frequency", a fraction of mutants in a population that carry particular mutations in a gene that confer resistance to an antibiotic (in the current case, polymyxin B), which was calculated from fluorescent signal intensities at a position of Sanger sequencing data

Reviewer #2:

The main result of this study is that certain combinations of antibiotics can reverse multidrug resistance. This is a potentially interesting observation that could, in principle, help in the design of rational drug usage strategies to fight clinically relevant pathogens. However, this observation has been previously reported using different bacteria and drug combinations and therefore, although methodologically sound, I believe this study is not conceptually novel.

We thank the reviewer for the supportive comments. The reversal of evolved antibiotic resistance was indeed one of the main results in our study. We are also aware that there are several publications reporting such evolutionary behaviour (e.g. [10.1073/pnas.1320886110](https://doi.org/10.1073/pnas.1320886110), [10.1016/j.ijantimicag.2012.02.007](https://doi.org/10.1016/j.ijantimicag.2012.02.007) and [10.1038/nchembio.2176](https://doi.org/10.1038/nchembio.2176)). However, we think that our theoretical model will be the main novel contribution to the field because of the reasons explained below, and the reversing experiments were conducted to demonstrate the predictive power of our multi-drug evolution model.

The reasons why we think our study is novel are:

- (1) Existing math model studies for bacterial antibiotic resistance evolution are designed for / based on particular drugs or drug combinations. In contrast, we developed a general model that can be applied to a wide range of antibiotic combinations, yet showing good agreements with the experimental results.**
- (2) The model can potentially be used to predict *any* multi-drug evolutions because the model is based on evolutionary patterns under single antibiotic stresses. This may offer a way to search for effective antibiotic combinations for cycling.**

To reflect this, we modified the text as follows:

Line 364-369:

As the model predictions of multi-drug resistance are based on evolutionary patterns under single drug stress, the model could potentially be used to predict other multi-drug evolutions where the data of single drug evolution are available. Thus, as single drug evolution experiments require fewer trials than combinatory trials, application of the model may offer a way to search for effective antibiotic combinations for cycling based on a small number of experiments.

Indeed, in the last couple of years a series of published papers have demonstrated, both theoretically and experimentally, that drug resistance can be controlled by appropriately cycling between different antibiotics, even in stringent scenarios where the drugs used are structurally similar (10.1371/journal.pone.0122283, 10.1371/journal.pcbi.1004493) or select for cross resistance (10.1371/journal.pbio.1002104). None of these papers has been cited in this manuscript.

Thank you for the reference. We have cited the papers in the introduction to provide the context of our work as follows:

Line 43-45:

So far several laboratory evolution studies based on collateral sensitivity have proved the effectiveness of antibiotic cycling with certain drug combinations and dosing regimens^{9,10,28,29}.

The continuous culture device constructed by the authors is very interesting, as it allows implementation of feedback controllers that dynamically manipulate the environmental conditions from real-time estimates of bacterial density. Previous studies have focused on implementing serial dilution protocols with daily transfers, and therefore imposing a temporal structure characterised by growth-dilution cycles, but also limiting the switching rate to periods longer than one day. Surprisingly, despite the possibilities offered by their experimental system, the authors choose to implement periodic cycles with daily switches because “shorter intervals tended to delay the development of the antibiotic resistance” (page 5). This is a strange reason to ignore rapid switching rates, in particular given that the main goal of the study is “to suppress the development of drug resistant mutants” (page 2).

As the reviewer mentioned here, we constructed the device to overcome the limitations of the conventional serial dilution method, especially to keep bacterial cultures in the exponentially growing regime which is crucial to achieve a well-controlled physiological state. In principle, it is possible to switch antibiotics with shorter periods (e.g. every 6 hours). However, faster switching could result in mixing of two antibiotics. As our main focus in this study is the effect of temporally segregated antibiotic stresses where we limit the minimum switching rate to be one day to avoid mixing of antibiotics. We are currently investigating the effect of mixed drug stresses using the same system, which we hope to report these results soon.

The mathematical model posed by the authors is based on positive autoregulation and double negative feedback loops that seems to be in agreement with the experimental data (Figure 2). There are some cases where the data and the model produce different results (which is expected in a simple, phenomenological model) and the possible sources of disagreement are discussed in the Supplementary Material (Supp. Fig. 6a and Fig 3a). I would suggest this discussion should be included in the main text.

The suggested part was moved to the main text (results section, mathematical modelling part. See Line 264-276).

An important assumption of the theoretical model is the existence of a fitness cost associated with resistance, considered by imposing the constraint that “the resistance level eventually saturates” (page 6). I understand that this is a parsimonious modelling assumption, although (as the authors mention) some drug resistance mutations are essentially cost-free and, even if they are costly, the cost can be readily compensated. In any case, an immediate consequence of this assumption is that a multidrug resistant subpopulation would present a lower fitness than single-drug resistant strain in single-drug environments,

Indeed, the fitness cost is a key factor in the modelling of bacterial drug resistance evolution as widely recognised from experimental studies (e.g. 10.1038/nrmicro2319), and it is also implicitly integrated in our model. An increased fitness cost of multi-drug resistance was introduced as negative feedback loop --- If a (sub)population develops resistance to drug A, then it suppresses the development of drug B resistance. The higher the drug A-resistance level is, the more the drug B-resistance is suppressed. This construction of the model effectively means that the pathway to multi-drug resistance tends to be inhibited (due a lower fitness).

and therefore it's not surprising that multidrug resistance can be reversed when only one antibiotic is deployed.

It is indeed not surprising that multi-drug resistance can be reversed from a theoretical model point of view. This is why we conducted the reversing experiments. However, what was surprising to us was that drug resistance of a bacterial population can be reversed even after 24 day exposure to antibiotic cycling (Fig.4a, although this is a case of single-drug resistance). In contrast, a population with 12 day exposure to single antibiotic (Supplementary Fig.8c) did not show any reversal. A possible explanation for these different outcomes was that antibiotic cycling prevented selective sweeps to happen. We also found that this genetic/phenotypic heterogeneity was a key for the reversibility of drug resistance. To our best knowledge, this heterogeneous population aspect was not recognised in the previous antibiotic cycling research (experimental/theoretical), and hence it is one of our contributions to the field. To reflect these points, we modified the text as follows

Line 298-305:

One of the important implications in the model is that the evolution of bacterial drug resistance can be viewed as a dynamical system because the model proposed here is a simple deterministic system. Such dynamical systems view of biological systems was previously proposed in the context of stem cell differentiation and drug resistance evolution of cancer cells^{44,45}. In the current context, this view suggests that drug resistant states of bacterial population can be directed from one state to another if antibiotic stresses are modulated externally. In fact, it was theoretically indicated

that antibiotic cycling with a pair of synergistic drugs can select susceptible bacteria while eliminating drug resistant ones from a population²⁰.

Line 340-344:

This result suggests that the cycling of antibiotics prevented selective sweep and thus maintained the genetic heterogeneity in the population even after 24 day of antibiotic exposure. This result was supported by the fact that the populations evolved under a single antibiotic stress did not show the reversal of drug resistance as the population was genetically homogeneous.

Actually, it has already been shown mathematically that in a 2-locus, 2-allele model of a continuous culture device where drug-resistance is costly, that a multidrug resistant subpopulation can be minimised by appropriately cycling antibiotics, even when the drugs used interact synergistically (10.1098/rsif.2012.0279).

Thank you for pointing to a relevant literature. We have cited the study in the result section. See the previous reply (Line 298-305, ref 20).

I have serious concerns about the sample preparation used for the single-clone profiling and whole-genome sequencing. The authors describe in their methods (page 16) that evolved strains were diluted in LB with no drug and grown overnight before DNA extraction and performing MIC assays. I understand the need for greater densities in order to obtain large DNA quantities necessary for Illumina MiSeq sequencing, but removing the selective pressure in favour of resistant mutants could have significant consequences on the MIC of the evolved population, as well as in the mutations identified from genome sequencing. This step is critical in particular when, as argued by the authors, drug-resistance mutations are associated with a fitness cost (see, for example, 10.1093/gbe/evu106).

The samples for MIC assay and genome extraction were prepared in the same method, overnight incubation in LB. Hence, all the MIC and genome data in the paper all are corresponding.

In addition, we tested if the resistance level changes during overnight incubation in the presence or absence of antibiotics. We used the resistant samples evolved under single antibiotic conditions and compared MIC values after incubated overnight with and without antibiotics. The MIC values were same regardless of antibiotic exposures during the overnight incubation (see Supplementary Fig. 11).

Furthermore, we measured colony forming unit of a resistant strain to see if the resistance level changes during overnight incubation without antibiotics. We used the final day sample of POL and CHL cycling (Fig 4a, 24th day) which had acquired only POL resistance through the cycling experiment. The frequency of survivors was compared between a frozen stock and overnight culture of the stock without antibiotics. Again, no significant difference was observed between two conditions (Supplementary Fig. 12).

These results show that the overnight incubation before MIC measurement or genome extraction did not affect to the MIC levels of samples.

Above explanation was added in the Method section (Line 487-495).

My main concern about this manuscript, however, is that the authors ignore a considerable body of literature relevant to the antibiotic cycling problem and yet they state in the introduction that “systematic investigations into how a bacterial population develops resistance to multiple cycling antibiotics have yet to be done”. Indeed, this is a very difficult

problem and one that needs more attention from scientific and medical communities and, although I think this study could be a contribution in the search for rational antibiotic deployment strategies

We feel there is a misunderstanding here regarding the term "systematic". With "systematic investigations", we do *not* mean investigations using theoretical models, but “exhaustive experimental investigation into bacterial evolutionary patterns by testing a number of antibiotic cycling combinations”, which we think it is true, to our best knowledge. We apologise for using a confusing term and now it is corrected as follows:

Line 45-47:

However, an exhaustive experimental investigation of effective antibiotic cycling combinations for suppressing the emergence of bacterial drug resistance is yet to be conducted.

I also believe its imperative that the authors put their findings in the context of previously published studies.

In addition to the corrections mentioned above, we expanded the introduction and discussion sections to describe our study in the context of previous studies. Please refer to Line 351-377 for details.

Reviewer #3:

*Yoshida and coworkers performed an extensive study in which they experimentally and theoretically elucidated how different temporal antibiotic dosing schemes affect the development of antibiotic resistance in *E. coli*. Motivated by recent experimental studies, they constructed a continuous-culture device, which enabled them to dynamically adjust drug concentrations both on short time scales (to ensure constant selection pressure) and on longer time scales (to switch between different antibiotics). This custom-built device that is similar to the “morbidosat” (Toprak et al., Nature Genetics, 2012) provides experimental flexibility and is well suited for assessing how the duration of antibiotic treatment prevents or promotes resistance evolution to sequential antibiotic treatments. The authors demonstrated that temporal cycling of drugs promotes genotypic and phenotypic heterogeneity, which is at the heart of resistance reversal. Their findings are supported by genotyping of bacterial populations and by a phenomenological model of resistance evolution in a multidrug environment. The experimental part of the study appears largely well done with clearly described methods (where these deviate from reported ones). Careful experimental and theoretical studies are needed to understand and combat antibiotic resistance evolution; hence, this study will likely be interesting for a broad readership. The generated dataset is quite comprehensive and will represent a good reference for further studies of antibiotic resistance evolution.*

However, I have several concerns (see below) that need to be clarified to corroborate the proposed claims and increase confidence in the reliability of the data.

We thank the reviewer for supportive comments and also for understanding the novelty of our work. We performed additional experiments and revised the manuscript according to the comments.

Major points:

1. The result that absolutely no resistance to one of the drugs evolves e.g. when KAN and POL or CHL and POL are switched would be quite striking but needs additional clarification. For one, conceptually similar experiments (using different drugs) were reported in the cited

reference (Kim et al., PNAS, 2014) but this study consistently found increases in resistance to both drugs (as one would generally expect). What causes this difference? Some plausible explanation for this would be important.

We made a section (“Evolutionary patterns and possible mechanisms of drug resistance suppression”) to explain the observed patterns. Simply put, we think this would be due to the different culture methods. The previous study by Kim et al. used the serial transfer method. In general, bacterial evolution would be slower with this method as the bacteria need to adapt to the log and stationary phase conditions during overnight culture, while the log phase is always maintained in morbidostat, which may have led to various evolutionary patterns (e.g. rapid evolution with KAN or NAL resistance, slow evolution with CHL or NIT resistance). Under antibiotic cycling conditions, evolution with serial transfer method would be even slower, and hence it showed monotonic increases consistently. In contrast, evolution in morbidostat would be diverse due to the combination of various single evolution patterns.

Please refer to the section (page 6) for more details.

Also, the data in Fig. 1c,e that show these results have features that require some explanation: in panel c, POL resistance increases by ~2-fold on day 10 but then decreases again on day 11 even though POL is present at that time. It would help to explain how that can happen because this selection against these first resistant mutants in the presence of the drug may be important for the observed phenomenon. In panel e, CHL resistance seems to start increasing right for the last data point on day 24.

We think this is hard to distinguish if it is the onset of drug resistance or just spontaneous fluctuation. In fact, similar phenomena were reported in the previous morbidostat paper that the resistance level decreased temporarily even in the presence of antibiotics (10.1038/ng.1034, Fig 2d and e). Such small increase/decrease in POL or CHL resistance were observed in the other cases (e.g. Supplementary Fig.S2a POL-KAN, and Fig.S3a CHL-POL) and this fluctuation was observed even when the antibiotics were not used (Supplementary Fig.S2a POL-KAN day 5 and 10).

A possible reason for such fluctuation would be heteroresistance and chemical communication of polymyxin resistance (cf. 10.1128/CMR.00058-14 for review), as mentioned above (Line 170-182). It is known highly POL resistant subpopulation release chemical signals that protect less resistant population from POL stress. If the size of highly resistant subpopulation fluctuates (e.g. when it was transferred to a new vial on a next day), the resistant level as the whole population may fluctuate as well.

Overall, it would greatly help to explain the plausibility of these results in the framework of basic population genetics (see below).

Please refer to the new section (“Evolutionary patterns and possible mechanisms of drug resistance suppression”), which we discuss the unique evolutionary patterns we observed, i.e. (1) the delayed POL resistance evolutionary trajectory, (2) suppression of POL resistance in KAN-POL cycling, and (3) suppression of CHL resistance in CHL-POL cycling. We tried to explain the behaviour from population genetics as well as population dynamics point of view.

2. The theoretical model described in the Supplement contains parameters that are closely related to the parameters of the logistic equation used to quantify the abruptness of the antibiotic resistance onset. Are there any analytical ways to connect the parameters L , k and m of the logistic equation with the parameters n , θ and α from the dynamical

system? It would help to reduce the number of parameters/equations to keep the text more concise and easier to follow.

Yes, L , k , and m of the logistic equation correspond to $1/k_A$, θ , and α_A , respectively. However, it should be noted that they correspond qualitatively, not quantitatively. We considered using logistic parameters for the model, but discarded the idea because of two reasons:

- (1) m and θ do not correspond directly because m is the mid-point of the curve while θ is the start point. They can be in principle converted using the definition of theta as $\theta = m - \frac{\log(10L-1)}{k}$ with $L = \frac{1}{k_A}$ and $k = \alpha_A$ (see "logistic fitting of experimental data" section in the methods), but this makes the model equation complex.
- (2) Even if the first term is replaced by the logistic function, theta is still used in the second term of the equation. Replacing the term with m will make the equation even more complex.

Thus we decided to use the current form of the equation as it is simple and symmetric, although new parameters were introduced. To make the text easier to follow, we added following text:

Line 235-239 in the Supplementary Information:

It should be noted that the above equation gives temporal evolutionary curves similar to logistic curves. Although the logistic function $f(x) = \frac{L}{1+e^{-k(x-m)}}$ can be correlated to the above equation by $L \sim \frac{1}{k_A}$, $k \sim \alpha_A$, and $\theta = m - \frac{\log(10L-1)}{k}$, we used the current form of the equation instead of the logistic equation because of the simplicity and symmetry with the second term as described below.

In the model, the Hill coefficients are first introduced as free parameters, but are later set to a constant value for all cases. The same applies for gamma, k_A and k_B ; how were these values determined? A quantitative assessment of the impact of parameter values should be provided.

We have added a new part to the supplementary text to explain how we determined the parameter values. Please see a section "Determining system parameters", Line 256-282 in the Supplementary Information.

Collateral sensitivity enters into the model through the coupling constant beta, which is given as $\beta = \gamma 2^{-x}$, where x is a value from the collateral sensitivity/resistance matrix in Fig. 6. Values in the matrix are log2 transformed MIC-fold changes; hence, beta directly equals gamma times the MIC fold change. However, the reported values for CHL-POL and POL-CHL do not agree with this (given that $\gamma = 0.1$).

Please note that negated value (-x) was used to calculate beta. Thus, beta is gamma times the reciprocal of the MIC fold change. For example, the collateral sensitivity/cross resistance values for POL and CHL are 0.768519 (rounded to 0.8) and -0.541667 (-0.5), thus beta values are calculated as

$$\beta_{\text{POL-CHL}} = 0.1 \times 2^{-0.768519} = 0.0587 \sim 0.06$$

$$\beta_{\text{CHL-POL}} = 0.1 \times 2^{0.541667} = 0.1455 \sim 0.15$$

We also noticed that the values in the matrix was rounded, but we used the original values for calculation (Line 252-255 in the Supplementary Information), hence they were not completely in agreement. We apologise for the error and provided the original data as Supplementary Data 2.

Setting all coupling constants beta to 0 might be a useful null-model when comparing correlation between the observed and predicted MIC of drugs (Fig. 3). Such a null model would be useful to assess how successful the theoretical predictions are.

We added scatter plots (similar to Fig.3) comparing a null model (with $\beta = 0$) with experimental data as Supplementary Fig. 10. Such null model has no interaction between antibiotic resistance and hence resistance levels to both antibiotics approach to the maximum. As a result, R^2 values were 0.03 and 0.20 for the two drug resistances, respectively, while 0.44 and 0.64 with our model. We modified the manuscript to reflect this. Please see Line 257-263.

It would help to emphasize that this model is phenomenological and it should be explained why the authors chose this model over well-established population genetics models for such evolution experiments

We agree with the reviewer. Our model presented here simulates bacterial evolution at the phenotypic level and no underlying genetic dynamics was incorporated. Although population genetics model, such as 10.1126/science.1122469 and 10.1371/journal.pbio.1002299 can be used to model the observed evolutionary behaviour, we here used the phenotypic evolutionary model because of reproducible phenotypic evolutionary patterns despite diverse underlying genotypic mutations in morbidostat, as previously observed by Toprak and colleagues. Similar evolutionary patterns were also observed in the bacterial ethanol stress adaptation by some of us, supported by a theoretical model. We modified the text as follows:

Line 204-210:

It should be noted that we employed a phenomenological model rather than commonly adopted population genetics models^{8,33,34}. Previous studies revealed that, despite diverse underlying genetic alterations, bacterial evolution for stress resistance can exhibit remarkably similar trajectories^{7,8,35-37}. Additionally, from a prediction point of view, it is helpful if a model is based on experimental parameters that can be measured relatively easily. Thus, we here developed a theoretical model relying only on relative MIC levels.

A population genetics model could provide deeper insight into the evolutionary dynamics, in particular the peculiar dynamics observed for POL (Fig. 1e): e.g. what is the beneficial mutation rate that would lead to such dynamics? The effect that POL resistance evolves perfectly reproducibly after exactly 7 days in this experiment is potentially interesting but deserves some explanation: is this plausible given the inherent stochasticity of evolutionary dynamics?

Although we do not have population genetics models to explain the observed phenomena at the moment, we are speculating that this reproducible dynamics of POL resistance evolution might be partly because of non-genetic drug resistance (modification to Lipid A and/or chemical communication, see 10.3389/fmicb.2014.00643 and 10.1128/CMR.00058-14 for reviews). A population can 'switch' to genetic resistance mechanisms (beneficial mutations) when the drug concentration exceeds a level that non-genetic mechanisms can cope. This might explain the step-like evolutionary patterns of POL resistance. See Line 169-182.

3. In the Discussion section, the authors argue: “From a clinical point of view, our finding is crucial because it suggests that the decreased efficacy of an antibiotic can be restored, even from a multi-drug resistant bacterial population, by adopting a time-modulated drug dosing strategy.” This claim may be true for de novo developed resistance in mixed populations and less so for clinically troublesome multi-drug resistant bacteria. Already the text on page 11 of the manuscript indicates that reversibility is possible but in a very special regime of a mixed population (which is maintained due to antibiotic cycling). These claims of clinical relevance should be supported by corresponding data or toned down.

We have corrected the corresponding part of the text to tone down the claim as follows:

Line 370-374:

Our findings indicate that the decreased efficacy of an antibiotic can be restored by modulating antibiotic stress however care must be taken when translating these results to clinical settings given that our experiments here considered de novo chromosomal mutations only. Further experiments, such as evolution experiments with plasmid-mediated antibiotic resistance, would be required to elucidate the feasibility of resistance suppression by antibiotic cycling in more practical situations.

4. In general, when stating and comparing quantities (e.g., k rates) errors and, where applicable, appropriate statistical tests of the claims should be reported. In order to maintain the flow of the text, the statistical tests used and the p -values could be reported in the supplement.

We added SEM values for k rates in the main text and in the Supplementary Fig. 5b. P -values were added in the main text and Supplementary Fig.4 (comparison of cycling rates). Details of the statistical measures were added in the Method section (“Morbidostat data analysis” section).

5. The cultures used in genome extractions were grown overnight in the absence of antibiotic. Was it verified that resistance occurrence in the population was not lowered (if associated with the fitness cost), e.g., by comparing the ratios of colony forming units on plates with and without antibiotic before and after overnight incubations?

The samples for MIC assay and genome extraction were prepared in the same method, overnight incubation in LB. Hence, all the MIC and genome data in the paper are corresponding.

We also performed additional experiments and measured the colony forming unit of a resistant strain to see if the resistance level changes during overnight incubation without antibiotics. We used the final day sample of POL and CHL cycling (Fig 4a, 24th day) which had acquired only POL resistance through the cycling experiment. The frequency of the survivors was compared between a frozen stock and overnight culture of the stock grown without antibiotics. However, no significant difference was observed between two conditions (Supplementary Fig. 12).

In addition, we tested if the resistance level changes during overnight incubation in the presence or absence of antibiotics. We used the resistant samples evolved under single antibiotic conditions and conducted two overnight incubations, with and without antibiotics. Again, the MIC values between these cases were quite close regardless of antibiotic exposures during the overnight incubation. The wild-type MIC level was used for the overnight incubation. These results show that the overnight

incubation before MIC measurement or genome extraction did not affect to the MIC levels of samples (Supplementary Fig. 11).

These results show that the overnight incubation before MIC measurement or genome extraction did not affect to the MIC levels of samples.

Above explanation was added in the Method section (Line 487-495).

6. In the morbidostat protocol that is used, what is the growth rate in the absence of the antibiotic?

The growth rate without antibiotics was ~1.0(/h), although it was constantly fluctuating due to measurement noise. Please see Supplementary Fig. 14.

What is the relative growth inhibition given the dilution and antibiotic concentration adjustment protocol described in the Method section? Is the relative inhibition indeed identical for all used antibiotics?

It was commonly observed that the growth rate fell between 0.2-0.4 after antibiotics were added (Supplementary Fig. 14). As the reviewer mentioned below, some antibiotics (e.g. KAN, NAL, AMP) indeed caused stronger inhibition than other antibiotics, which decreased the growth rate to below zero. However, the rate came back to 0.2-0.4 after the drug concentration in the cell culture was diluted by LB without antibiotics.

What is the average number of antibiotic concentration adjustments needed to achieve constant inhibition? Is this time comparable with the switching rate?

The average number of antibiotic adjustments depended on the antibiotics used. In general, the antibiotics with larger α (e.g. KAN) required more adjustments than those with smaller α (e.g. CHL). Please refer to Supplementary Fig. 15. We did not see any correlation between the number of adjustments and switching rate.

7. Some of the chosen antibiotics (kanamycin, ampicillin) are known to have a very steep dose-response curve, i.e. the growth rate as a function of antibiotic concentration has a very abrupt drop, which in turn makes it hard to control growth inhibition in a device like the morbidostat. How was this problem solved?

This is a reason why we used a higher OD threshold (OD=0.4) than that of the original morbidostat experiment (OD=0.15). If the threshold is low, the entire bacterial population could be killed by an antibiotic before the antibiotic concentration was diluted by the growth medium (LB), and the bacterial growth did not recover. Thus, we hypothesised that a proportion of bacteria would survive if the cell density is relatively high. Some antibiotics (e.g. KAN, AMP, NAL) indeed showed strong and abrupt growth inhibition as shown in Supplementary Fig.14. However, with a higher OD threshold, we observed the growth recovery after several hours of no growth period at very low OD (below the sensitivity of optical sensor), and were able to obtain the reproducible evolutionary patterns in the cases tested here.

We inserted following text to mention about this in the manuscript:

Line 433-435:

Note that we used a higher OD threshold than the previous study (OD_{THR}=0.15)⁷. This is because some antibiotics caused strong growth inhibitions than the other antibiotics (Supplementary Fig. 14), which make the control of growth inhibition difficult in morbidostat.

Sample growth curves for the single antibiotic regime for all used antibiotics should be provided (i.e. Supp. Fig. 1b with real data).

We added the data as Supplementary Fig. 14.

8. Figure 1 suggests that the MIC of CHL and NIT increased roughly 16 fold over the course of 12 days. Data from the cited references (Toprak et al., Nature Genetics, 2012) and (Chevereau et al., PLoS Biology, 2015) shows that the IC50 increased by a considerably larger factor for both drugs in similar experiments. It would be helpful to explain the (potential) origin of these discrepancies.

This would be because the total duration of antibiotics exposure was different from the previous studies. In our experiments, we exposed the bacteria to each antibiotic only for 12 days which is almost half the period compared to the other two papers. We compared the resistance levels at day 12 in the table below. For CHL resistance in our experiment, $\log_2(\text{MIC fold change})$ was 4.64 (Fig.1d), i.e. 25.4-fold increase. Cheverau et.al. showed between 20 to 65 times increase with 6 parallel experiment. For NIT resistance, we observed around 30 fold increase and Cheverau et.al. demonstrated around 20 times increase. Considering that we use different medium (LB) compared to the other two papers (M9 medium), we think this difference is not significant.

antibiotics	exposure days	Fold increase		
		Toprak et al.	Chevereau et al.	Our result
CHL	12	~ 230	~ 20 - 65	25.4
	22	~ 780	~ 200	-
NIT	12	-	~ 20	30.2
	22	-	~ 40	-

Supplementary Figure 5 states that the original data is shown in Fig. 1, which is not the case for RIF, AMP and TET. It would be important to provide the absolute increases in MIC for all drugs for cross-validation with existing literature.

The original data was added as Supplementary Fig.5b.

9. Collateral sensitivity/resistance profiles were measured before (e.g. in the cited reference Imamovic and Sommer, Sci. Trans. Med. 2013 and in Oz et al., Mol Biol Evol, 2014). It would be helpful to provide a scatter-plot comparing the measured values with the reference values.

We added a scatter-plot comparing the \log_2 -transformed relative MICs to the parent strain in Supplementary Fig. 13. We compared the four datasets here, one from Imamovic et al., two from Oz et al. (strong selection and mild selection), and ours. Generally, our dataset displayed agreements with other datasets.

These previous studies established that resistance to aminoglycosides (e.g. KAN) renders strains very sensitive to many antibiotics of other classes; this is less clear in the data presented here and it would be helpful to comment on this.

We also observed collateral sensitivities of KAN-resistant strain to other antibiotics (POL, CHL, NAL). However, it was not as evident as those of POL-resistant strain. We think this less clear KAN collateral sensitivity in our case was due to different

selection strength as shown by Oz et al. that different selection strength lead to different collateral sensitivity profile.

We inserted following text in the manuscript:

Line 140-144:

While these results were in general consistent with previous studies on collateral sensitivity (Supplementary Fig. 13)^{26,32}, some collateral sensitivity profiles (e.g. KAN) were less evident compared to previous cases. This may be due to different selection strengths during evolution between previous and our cases because the collateral sensitivity/cross resistance profile is known to be dependent on the selection strength³².

Minor points:

1. In the second paragraph of the introduction, it is worth mentioning that keeping bacterial cultures in the exponentially growing regime is crucial to achieve a well-controlled physiological state.

Corrected as follows:

Line 56-59:

The use of the automated cell culture system was important not only in delivering programmable dosing, but the morbidostat system enabled maintenance of bacterial populations in exponential growth phase. Thus the platform was expected to provide good phenotypic reproducibility between parallel experiments¹⁴ whilst enabling genetically diverse populations to co-exist^{30,31}.

2. If x and m in the logistic equation are given in the units of time (e.g., days) then values for k should be reported with units of inverse time which is not the case. If some normalization was used that renders values unit-less, this should be reported.

"day⁻¹" was added as the unit for the parameter k .

3. In the section "Evolution against single antibiotic ..." evolutionary patterns are divided into two categories according to the rapidness and length of lag phase: were the k and m values used as a classifier? If so, the threshold values should be reported.

We used k and θ (the length of lag phase) as classifiers with threshold $k = \theta = 2.5$. θ was calculated as a time when the resistance level exceeds 0.1 using logistic function and the parameter values k and m and L (maximum value of the curve) as
$$\theta = m - \frac{\log(10L-1)}{k}$$
. We added calculated values in Supplementary Fig.S5b. The equation to calculate θ is added in Methods (Line 476-477).

4. Comparison of the experimental results in Fig. 3 should be done more quantitatively by reporting the corresponding correlation coefficients or other metrics.

R^2 values were added in Fig.3c and d.

Also, figure 3 should be explained in a bit more detail; including a concise explanation why pairs of cycled drugs appear twice in inverse order would facilitate the understanding of this important figure. Also, why are these pairs sometimes very similar (e.g. NIT-CHL, CHL-NIT) and sometimes extremely different in these plots (KAN-POL, POL-KAN)?

In Fig.3a and b, the normalised drug resistance level for one of the cycled antibiotics was plotted against that for another antibiotic. For example, a case of KAN-POL

cycling (KAN as drug #1 and POL as drug #2) was mapped on the x axis because only KAN resistance was developed while POL resistance was completely suppressed. In general, the results can be categorised into two groups, i.e. the single and multi-drug resistant states, as observed above. The former cases (e.g. KAN-POL) were plotted either on one of the axes, while the latter ones (e.g. NAL-KAN) were in the upper right part of the plot. To illustrate the predictability of the theoretical model, the normalised predicted MIC values were plotted against the normalised observed MIC values (Fig.3c and d). Overall, the model predictions showed good correlations with the experimental results ($R^2 = 0.44$ and 0.64 , respectively).

We added these details in Fig.3 caption, Line 287-296.

5. Morbidostat data analysis: Is the expression for the growth rate missing a logarithmic transformation of OD values? If the written equation was actually used, please report the regime in which such linear approximation is valid.

We apologise for the typo. The equation is now corrected as $\mu = \log\left(\frac{OD_n^E}{OD_n^S}\right) / \Delta t \times 3600$.

6. Supplementary table 1: Chloramphenicol acts on the 50S ribosomal subunit, more precisely on peptidyl transferase center, not on 30S ribosomal subunit as reported in this table.

The information in the table was corrected.

7. There are some language errors (a few examples are listed below) that should be corrected throughout:

- a. Supp.Fig.6: Blue and red grids indicate that the antibiotic pair is collateral[ly] sensitive and cross resistance [resistant?], respectively.*
- b. Supp.text, p 17: "absent (i.e. = 0), the resisntance"*
- c. Main text p 14: "After autoclave [autoclaving?], silicone and PTF"*
- d. Main text p 16 "96 well plate at at 30 °C, we remove[d]"*

Corrected. We also checked the manuscript for other grammatical errors very carefully.

REVIEWERS' COMMENTS:

Reviewer #1 (Remarks to the Author):

GENERAL COMMENTS

I believe this will be an important paper that deserves to be written very carefully. Below I have made a few suggestions that may improve comprehension speed and author credibility. These comments are not comprehensive. My major comment concerns the Discussion (marked with an asterisk below). It should be rewritten.

SPECIFIC COMMENTS

Line 26 sentence regarding cycling. This has been tested clinically with mixed results and is not a general solution to the problem (squeezing the balloon). Readers interested in the present work will know that, and they will want a bit more comment and reference citations to establish the knowledge base of the present authors regarding the clinical situation. This is addressed somewhat at line 38, but that seems a bit late.

Line 32. Strange comma use

Line 38. due to conflict. I suspect that if cycling worked, it would be in use without laboratory or mathematical support, as this field tends to be empirically driven. Thus, I suggest revising this sentence to remove the causality

Line 40 and many places elsewhere. Compound adjectives require commas to maintain reading speed. Another example is on line 135.

Line 41 collateral sensitivity. Please define so the reader does not have to refer to the cited literature. A few sentences later the term takes on major importance. Remember that you have a general audience.

Line 47 paragraph above. So what is the fundamental idea? Is it that there are fitness costs associated with resistance and that the cycling allows the wild type for a particular resistance gene to overgrow? If so, you can weave that, or another principle, into the paragraph and give the exercise biological foundation.

Line 48 fundamental. I would delete this pretentious word. It is unnecessary.

Line 49 long-term. Add in parentheses the approximate length. Every reader has a different definition, which makes the present usage meaningless.

Line 54 antibiotic resistance. Readers interested in this work will use the clinical definition of resistance, one that is determined by breakpoints. You are actually measuring changes in susceptibility. While your meaning is clear, there is no need to be imprecise. At line 437 you define resistance in a very simple way. I suggest that you move that definition to the text to make it clear that you are not being sloppy with language.

Line 97 dynamic antibiotic concentrations. Please indicate where these are defined. I assume you mean fluctuating as in a pharmacodynamic model, but that is not at all clear. At this point the reader will want to know drug concentrations relative to MIC. It turns out that the information is in Methods, but it is too important to be hidden there.

Line 70. Isogenic. What do you mean here? Your methods say that you simply used this strain. Isogenic with reference to what? Do you really mean that the sample was streaked to single colonies and one colony was selected and saved for all experiments?

Line 73. Avoid cross-resistance. Even with different classes you may not be able to avoid efflux- or uptake-based resistance. A solution is to qualify the sentence by saying target-

based resistance. Do your high concentrations (10x MIC) make it unlikely to see efflux? What does your sequence analysis show? These are important considerations in word choice at this point.

Line 128. As such comma

Line 125. Evolutionary patterns. Here and in many other places you use phrases that are likely to be unfamiliar to some readers. This immediately halts reading and understanding, even though in some cases you later define the terms. If you do not want to define a term at its first use, add in parentheses “defined below”.

Line 172. Multiple mutations are required ... This is a spot where the reader expects the work to have addressed this question with nucleotide sequence information. Since you later give this information, indicate that here as described below – don’t make the reader think that you left a big hole in the study.

Line 176. Please explain why heteroresistance is not in the category of genetic mutation. By genetic mutation I assume you mean DNA-based mutation. The logic and writing of this passage needs to be more careful. You may be able to solve some of the problem by using “new spontaneous mutation”. Are there mutations in *E. coli* that are not genetic, that is, DNA based?

Line 180. Low and high drug concentrations. To this point I have not seen any information on drug concentration. Thus, the meaning is unclear. You have that information in the methods, but it breaks up the flow to have to flip to methods to understand the general nature of the experiment. Moreover, this important point is hidden in a section of the Methods.

Line 194. Loss of resistance. My initial comment was that this seems like an important point and the data should be in the main text, not the supplementary material. Then later I find a section dealing with this topic. The solution is a “discussed below” statement. This example raises a general problem in the manuscript: putting the cart before the horse. I suggest that the entire manuscript be carefully reviewed so other examples can be corrected.

Line 309. Please add a reference for last resort.

Line 358. Delete exhaustive. This is a red flag to the reader. In general, avoid value-judgment statements about your own work. Let the reader draw this conclusion.

Discussion. This section adds very little to the paper and would be better titled “Summary”. A Discussion would delete the summary material and focus on 1) how your results differ from failed clinical and animal studies, 2) how your studies differ from pharmacodynamic models (see Firsov and Zinner) and how the two can be combined to be closer to reality, and 3) how your studies can be extended to animal systems (examples are tissue-cage studies in which cycling can be performed along with sampling of bacterial populations and drug concentration [see Cui 2006 *J. Inf Disease*]). Such a treatment will excite experts in other aspects of resistance to use your findings, because they will not have to think so hard to get started.

*Line 437. The drug concentration information is hidden in the Methods. This very simple statement should be early in the text, because concentration is extremely important for the entire work. For most readers, using concentrations at 10-times MIC constitutes extreme stress, in most cases lethal stress. More comment on concentration is required. In particular, it would be good to state (in the Discussion) that while you measured MIC, it is likely that the concentrations you used were actually lethal (for the lethal drugs) and that further work is needed to assess the importance of killing (this will make it clear that you understand the difference between assays for blocking growth and assay for survival [killing]). You may also want to mention that 10x MIC is likely to place concentrations high in the mutant selection window (depending on the drug), which will limit the resistance alleles you expect to be

selectively enriched (such a statement will make it clear that you understand the importance of concentration; a citation may be a review in Clin Inf Dis (2007) 44: 681 -688 by Drlica and Zhao).

Reviewer #2 (Remarks to the Author):

The revised manuscript clarifies many of the issues raised by the reviewers and implemented the suggestions of my previous report in a fully satisfactory manner.

Reviewer #3 (Remarks to the Author):

In the revised version, the authors have carried out additional experiments and analysis and improved the text of the manuscript. They have addressed virtually all my concerns adequately. I still have doubts about choosing the presented phenomenological model over established population genetics models, but at least this is clearly stated and the analysis of the former model appears sound. I also appreciate the addition of statistical measures of significance and appropriate tests. Overall, I recommend the manuscript for publication.

A minor language point: I would try to think of a new term for the time needed before the onset of resistance (currently "lag phase"). The reason for that is the newly added explanation of the discrepancies between previous evolution experiments with serial transfer methods, where stationary phase is the main speculative reason for the observed disagreements (although "lag phase" is not referred to directly).

Responses to the reviewer comments:

We thank the reviewers for detailed suggestions to our manuscript. We have answered all the comments below. As a result, we feel our manuscript was greatly improved.

Reviewer #1 (Remarks to the Author):

GENERAL COMMENTS

I believe this will be an important paper that deserves to be written very carefully. Below I have made a few suggestions that may improve comprehension speed and author credibility. These comments are not comprehensive. My major comment concerns the Discussion (marked with an asterisk below). It should be rewritten.

We thank the reviewer for the detailed comments. We have revised and expanded our manuscript. Please refer to our responses below.

SPECIFIC COMMENTS

Line 26 sentence regarding cycling. This has been tested clinically with mixed results and is not a general solution to the problem (squeezing the balloon). Readers interested in the present work will know that, and they will want a bit more comment and reference citations to establish the knowledge base of the present authors regarding the clinical situation. This is addressed somewhat at line 38, but that seems a bit late.

We added/modified related sentences to describe clinical studies on antibiotic cycling as follows:

L 37-43:

Antibiotic cycling has also been studied in clinical settings for over 30 years, particularly cycling of aminoglycosides was widely studied due to increasing drug resistance mediated by plasmids carrying aminoglycoside enzymes^{21,22}. Cycling in the intensive care unit was also actively investigated because infections with drug-resistant bacteria can be lethal if treatment fails²³.

Despite a number of clinical, laboratory, and theoretical studies with various antibiotic combinations^{20, 24-28}, they showed mixed results partly because of a lack of standard procedures to perform experiments.

Line 32. Strange comma use

Removed.

Line38. due to conflict. I suspect that if cycling worked, it would be in use without laboratory or mathematical support, as this field tends to be empirically driven. Thus, I suggest revising this sentence to remove the causality

We modified/deleted the sentences. Please see our response for Line 26 above.

Line 40 and many places elsewhere. Compound adjectives require commas to maintain reading speed. Another example is on line 135.

All corrected.

Line 41 collateral sensitivity. Please define so the reader does not have to refer to the cited literature. A few sentences later the term takes on major importance. Remember that you have a general audience.

We added a short explanation of the phenomenon as follows:

L 44-46:

However, recent studies suggest that exploitation of collateral sensitivity, in which bacterial strains resistant to an antibiotic exhibit increased susceptibility to other antibiotics^{12,13,29}, may be key to the suppression or reversal of the evolution of bacterial drug resistance³⁰.

Line 47 paragraph above. So what is the fundamental idea? Is it that there are fitness costs associated with resistance and that the cycling allows the wild type for a particular resistance gene to overgrow? If so, you can weave that, or another principle, into the paragraph and give the exercise biological foundation.

Yes, as the reviewer said, the fundamental idea was to exploit increased fitness cost incurred by collateral sensitivity and selectively kill resistant strains. We expanded following sentence to include the idea:

L 46-49:

Thus temporally modulated use of different antibiotics is a promising candidate method for effective multidrug dosing because switching of antibiotics can selectively perish resistant strains due to fitness costs associated with drug resistance while allowing susceptible strains to overgrow.

Line 48 fundamental. I would delete this pretentious word. It is unnecessary.

Deleted.

Line 49 long-term. Add in parentheses the approximate length. Every reader has a different definition, which makes the present usage meaningless.

Changed to "long-term (24 days)".

Line 54 antibiotic resistance. Readers interested in this work will use the clinical definition of resistance, one that is determined by breakpoints. You are actually measuring changes in susceptibility. While your meaning is clear, there is no need to be imprecise. At line 437 you

define resistance in a very simple way. I suggest that you move that definition to the text to make it clear that you are not being sloppy with language.

We changed it to "the development of antibiotic resistance (measured as changes in susceptibility)" (L 69-70) and also moved our definition to the end of the paragraph from Method section.

L 83-86:

Note that we here defined a wild-type *Escherichia coli* MG1655 strain as a susceptible strain, in contrast to the clinical definition in which a strain can be treated with an antibiotic at the recommended dosage. Resistant strains were defined as any (evolved) bacterial samples whose MICs exceed that of the wild-type strain.

Line 97 dynamic antibiotic concentrations. Please indicate where these are defined. I assume you mean fluctuating as in a pharmacodynamic model, but that is not at all clear. At this point the reader will want to know drug concentrations relative to MIC. It turns out that the information is in Methods, but it is too important to be hidden there.

We added another sentence to include a detailed explanation of the morbidostat and the drug concentration. Please note that the morbidostat system tries to maintain the mutant selection window. Namely, when the growth medium containing 10xMIC antibiotic is administered to a bacterial culture, the antibiotic will be diluted 1/12 times. As described below, antibiotic is added at least a few times and thus the antibiotic concentration in the culture will be above MIC, but much lower than 10xMIC (cf. Supplementary Fig.14). Please also note that the paragraph was moved to Results section.

L 71-79:

In the morbidostat, the antibiotic concentration was automatically controlled by a custom algorithm⁷; when a bacterial culture reaches a certain density, one mL of a growth medium containing an antibiotic was administered into twelve mL of a bacterial culture, while a medium containing no antibiotic was added at low cell density (Supplementary Methods). The antibiotic concentration in the administered growth medium was at least ten times higher than the minimum inhibitory concentration (MIC) of a wild-type strain. Typically, antibiotics were added at least a few times once the cell density exceeds a threshold level, and thus the antibiotic concentration was increased to the range of drug concentrations that select for drug-resistant mutants (so-called "mutant-selection window"³³).

Line 70. Isogenic. What do you mean here? Your methods say that you simply used this strain. Isogenic with reference to what? Do you really mean that the sample was streaked to single colonies and one colony was selected and saved for all experiments?

Yes, by isogenic, we meant the strain originated from a single colony of *E.coli* MG1655 strain. We added this explanation in the Method as follows:

L 367-369:

***Escherichia coli* strain MG1655 was purchased from DSMZ (Germany) and streaked to single colonies. One single colony was selected and used throughout this study.**

Line 73. Avoid cross-resistance. Even with different classes you may not be able to avoid efflux- or uptake-based resistance. A solution is to qualify the sentence by saying target-based resistance. Do your high concentrations (10x MIC) make it unlikely to see efflux? What does your sequence analysis show? These are important considerations in word choice at this point.

We amended the sentence to emphasise that the chosen antibiotics to have different targets, and also changed the word "avoid" to "minimise" as weak cross resistances were commonly observed (Fig. 2).

L 89-91:

Antibiotics were chosen from different classes so that they have different targets (Table 1), which minimise the risk of developing potential cross resistance between antibiotics^{12,13,29}.

Regarding sequence analysis, we saw only one mutation in the efflux systems (marR gene in CHL and NIT cycling), although a few mutations in the upstream of mdfA (a multi-drug efflux system) were observed. Thus, we think the effect of multi-drug resistance by efflux pump mutations are negligible.

Line 128. As such comma

Corrected.

Line 125. Evolutionary patterns. Here and in many other places you use phrases that are likely to be unfamiliar to some readers. This immediately halts reading and understanding, even though in some cases you later define the terms. If you do not want to define a term at its first use, add in parentheses "defined below".

We added the definition of evolutionary pattern at its first use as follows:

L 56-58:

We found unique evolutionary patterns (i.e. temporal development of bacterial drug resistance) in which the emergence of drug resistance to one of the cycled antibiotics was completely suppressed

Line 172. Multiple mutations are required ... This is a spot where the reader expects the work to have addressed this question with nucleotide sequence information. Since you later give this information, indicate that here as described below – don't make the reader think that you left a big hole in the study.

We added a sentence to point the readers to the relevant genome sequencing data:

L 176-177:

In addition, as described below, the whole genome sequence data did not support the explanation (Supplementary Table 1).

Line 176. Please explain why heteroresistance is not in the category of genetic mutation. By genetic mutation I assume you mean DNA-based mutation. The logic and writing of this

passage needs to be more careful. You may be able to solve some of the problem by using “new spontaneous mutation”. Are there mutations in E. coli that are not genetic, that is, DNA based?

Heteroresistance is a phenomenon that a seemingly isogenic bacterial population shows various levels of susceptibilities to an antibiotic. In the case of polymyxin resistance, it was due to different gene expression levels of certain proteins. Thus no DNA-based mutations are involved here. We modified the sentence to make this point clearer:

L 179-183:

Recent studies indicated that heteroresistance to POL was achieved by upregulations of putrescine synthesis and YceI protein without any genetic mutations. In addition, release of the molecules from highly resistant subpopulations of heteroresistant bacteria protects less resistant bacteria from POL and other antibiotic stresses^{39,40}.

Line 180. Low and high drug concentrations. To this point I have not seen any information on drug concentration. Thus, the meaning is unclear. You have that information in the methods, but it breaks up the flow to have to flip to methods to understand the general nature of the experiment. Moreover, this important point is hidden in a section of the Methods.

We have added the information on the antibiotic concentration in the first paragraph of Results section. See our response above (Line 97)

Line 194. Loss of resistance. My initial comment was that this seems like an important point and the data should be in the main text, not the supplementary material. Then later I find a section dealing with this topic. The solution is a “discussed below” statement. This example raises a general problem in the manuscript: putting the cart before the horse. I suggest that the entire manuscript be carefully reviewed so other examples can be corrected.

We added a pointer to the later section as suggested:

L 198-200:

The latter case of CHL-POL cycling, in which CHL resistance was completely suppressed, can be explained by the reversible nature of CHL resistance (which was also observed with POL resistance, as described below).

Line 309. Please add a reference for last resort.

Added reference to a review paper on polymyxin resistance mechanisms.

Line 358. Delete exhaustive. This is a red flag to the reader. In general, avoid value-judgment statements about your own work. Let the reader draw this conclusion.

Removed.

Discussion. This section adds very little to the paper and would be better titled “Summary”. A Discussion would delete the summary material and focus on 1) how your results differ from

failed clinical and animal studies, 2) how your studies differ from pharmacodynamic models (see Firsov and Zinner) and how the two can be combined to be closer to reality, and 3) how your studies can be extended to animal systems (examples are tissue-cage studies in which cycling can be performed along with sampling of bacterial populations and drug concentration [see Cui 2006 J. Inf Disease]). Such a treatment will excite experts in other aspects of resistance to use your findings, because they will not have to think so hard to get started.

Thank you for the suggestions. We expanded the discussion to include possible experiments as suggested as follows:

L 333-336:

Although collateral sensitivity has been known for more than 60 years⁵³, the importance of this phenomenon for suppressing the development of drug resistance was not well recognised until recently⁵⁴, which may account for failed clinical or animal studies on antibiotic cycling.

L 353-364:

Evolution experiments with different modes of dosing with antibiotic cycling would be another possible route to better understand bacterial antibiotic resistance. In contrast to the morbidostat system where mutant selection is the main focus, *in vitro* dynamic model simulates pharmacokinetic profiles of antibiotics⁵⁵, which can provide useful insights into the effect of antibiotic cycling, particularly collateral sensitive pairs, and how drug resistance may develop *in vivo*. Furthermore, antibiotic cycling experiments exploiting collateral sensitivity can also be extended to animal models. Tissue cage model⁵⁶, for example, would allow constant sampling of bacterial populations as well as drug concentration during alternating antibiotics treatments. In parallel with these possible further experiments, we foresee that deeper understanding of population dynamics under multi-drug conditions and improved theoretical modelling would potentially eliminate labour-intensive combinatorial experiments with multiple drugs. Taken together, they may lead to novel therapies that reverse bacterial multi-drug resistance.

** Line 437. The drug concentration information is hidden in the Methods. This very simple statement should be early in the text, because concentration is extremely important for the entire work. For most readers, using concentrations at 10-times MIC constitutes extreme stress, in most cases lethal stress. More comment on concentration is required. In particular, it would be good to state (in the Discussion) that while you measured MIC, it is likely that the concentrations you used were actually lethal (for the lethal drugs) and that further work is needed to assess the importance of killing (this will make it clear that you understand the difference between assays for blocking growth and assay for survival [killing]). You may also want to mention that 10x MIC is likely to place concentrations high in the mutant selection window (depending on the drug), which will limit the resistance alleles you expect to be selectively enriched (such a statement will make it clear that you understand the importance of concentration; a citation may be a review in Clin Inf Dis (2007) 44: 681-688 by Drlica and Zhao).*

As we noted above, we mentioned the drug concentration in details that the drug concentration was above MIC, but much lower than 10xMIC because added antibiotic was diluted in the cell culture. We cited Drlica and Zhao as ref 33.

Reviewer #2 (Remarks to the Author):

The revised manuscript clarifies many of the issues raised by the reviewers and implemented the suggestions of my previous report in a fully satisfactory manner.

We would like to thank Reviewer #2 again for valuable comments to our manuscript.

Reviewer #3 (Remarks to the Author):

In the revised version, the authors have carried out additional experiments and analysis and improved the text of the manuscript. They have addressed virtually all my concerns adequately. I still have doubts about choosing the presented phenomenological model over established population genetics models, but at least this is clearly stated and the analysis of the former model appears sound. I also appreciate the addition of statistical measures of significance and appropriate tests. Overall, I recommend the manuscript for publication.

We would like to thank Reviewer #3 for very detailed comments, which greatly helped improve our manuscript.

A minor language point: I would try to think of a new term for the time needed before the onset of resistance (currently "lag phase"). The reason for that is the newly added explanation of the discrepancies between previous evolution experiments with serial transfer methods, where stationary phase is the main speculative reason for the observed disagreements (although "lag phase" is not referred to directly).

We agree with the reviewer. We changed "lag phase" to "silent phase" to avoid confusion with growth lag phase.